JCB Journal of Cell Biology

## TOOLS

# hGRAD: A versatile "one-fits-all" system to acutely deplete RNA binding proteins from condensates

Benjamin Arnold[1]*, Ricarda J. Riegger[1]*, Ellen Kazumi Okuda[1,2], Irena Slišković[1], Mario Keller[1,3], Cem Bakisoglu[1,3], François McNicoll[1], Kathi Zarnack[1,3], and Michaela Müller-McNicoll[1]

**Nuclear RNA binding proteins (RBPs) are difficult to study because they often belong to large protein families and form extensive networks of auto- and crossregulation. They are highly abundant and many localize to condensates with a slow turnover, requiring long depletion times or knockouts that cannot distinguish between direct and indirect or compensatory effects. Here, we developed a system that is optimized for the rapid degradation of nuclear RBPs, called hGRAD. It comes as a "one-fits-all" plasmid, and integration into any cell line with endogenously GFP-tagged proteins allows for an inducible, rapid, and complete knockdown. We show that the nuclear RBPs SRSF3, SRSF5, SRRM2, and NONO are completely cleared from nuclear speckles and paraspeckles within 2 h. hGRAD works in various cell types, is more efficient than previous methods, and does not require the expression of exogenous ubiquitin ligases. Combining SRSF5 hGRAD degradation with Nascent-seq uncovered transient transcript changes, compensatory mechanisms, and an effect of SRSF5 on transcript stability.**

## Introduction

RNA binding proteins (RBPs) often belong to larger protein families and form dense regulatory networks of auto- and crossregulation, wherein they complement or antagonize each other's functions. These networks ensure the robustness of gene expression starting from transcription in the nucleus to translation and mRNA decay in the cytoplasm. This renders it very challenging to disentangle the specific functions of individual RBPs and their link to human diseases (Gebauer et al., 2021; Müller-McNicoll et al., 2019). Most RBPs are also highly abundant and stable proteins, and many of them localize to nuclear condensates with a slow turnover, which makes it difficult to deplete them using RNA interference (RNAi). On the other hand, complete knockouts or very long depletion times trigger compensatory effects and do not allow to distinguish between direct and indirect effects.

One example of such a large family of nuclear RBPs is the serine/arginine-rich splicing factors (SRSF1–SRSF12, SR proteins) that are present in all metazoans (Sliskovic et al., 2022). SR proteins are essential factors for constitutive and alternative pre-mRNA splicing (AS), and they also influence many other steps in mRNA metabolism including transcription, alternative polyadenylation, mRNA packaging, export, translation, and cytoplasmic decay (Jeong, 2017; Wegener and Müller-McNicoll, 2019; Änkö, 2014). SR proteins also participate in the processing of non-coding RNAs and the formation and dynamics of nuclear compartments (de Oliveira Freitas Machado et al., 2023; Königs

et al., 2020; Wagner and Frye, 2021). In contrast to their canonical functions in splicing, the role of individual SR protein family members in specific AS events or other disease-related, non-canonical functions remains poorly understood. This is in part because SR and SR-like proteins cooperate and compete for binding sites, compensate for each other, and the removal of one SR protein can affect the levels of several other family members (Meinke et al., 2020; Änkö et al., 2012). Dysregulated levels of individual SR protein family members are associated with various cancers and other diseases including neurological disorders, liver disease, as well as coronary and cardiac diseases (More and Kumar, 2020; Ratnadiwakara et al., 2018; Zheng et al., 2020). Because of their overlapping functions and compensatory mechanisms, specific inducible protein degradation systems with short depletion times are needed to disentangle the functions of individual SR proteins and other RBPs.

Proteins are usually targeted for degradation by the addition of polyubiquitin chains to lysine residues which are recognized by the 26S proteasome (Komander and Rape, 2012; Yu and Matouschek, 2017). In a highly regulated, multienzymatic reaction, ubiquitin (Ub) is first activated by E1 enzymes under ATP consumption, subsequently transferred to E2-Ub-conjugation enzymes, and finally transferred to target proteins by E3-Ub ligases. More than 600 different E3-Ub ligases are encoded in the human genome, enabling target specificity (Nakayama and Nakayama, 2006). The RING-finger family contains the

[1]Institute of Molecular Biosciences, Goethe University Frankfurt, Frankfurt am Main, Germany; [2]International Max Planck Research School for Cellular Biophysics, Frankfurt am Main, Germany; [3]Buchmann Institute for Molecular Life Sciences, Goethe University Frankfurt, Frankfurt am Main, Germany.

*B. Arnold and R.J. Riegger contributed equally to this paper. Correspondence to Michaela Müller-McNicoll: mueller-mcnicoll@bio.uni-frankfurt.de.



SKP1-Cullin-F-box protein (SCF) complex, which is the most prominent E3-Ub ligase in mammals (Frescas and Pagano, 2008; Schulman et al., 2000). Its subunit RBX1 facilitates E2-Ub recruitment and the ubiquitylation reaction, CUL1 functions as a bridge, and SKP1 mediates binding to a modular F-box protein (FBP). More than 70 different FBPs are encoded in the human genome, providing additional target specificity (Frescas and Pagano, 2008; Schulman et al., 2000). FBPs have two major domains: the F-box domain, which is essential for the interaction with SKP1, and a variable C-terminal domain, which is essential for target recognition (Kipreos and Pagano, 2000).

Most targeted protein degradation approaches manipulate the SCF complex by introducing an engineered FBP with specificity for a protein-of-interest (POI). Examples are the deGradFP system (Caussinus et al., 2011) and the auxin-inducible degron (AID) systems (Li et al., 2019; Nishimura et al., 2009). The deGradFP system was developed to degrade green fluorescent protein (GFP)–tagged proteins in *Drosophila melanogaster*. Here, the substrate recognition of the F-box protein Slmb was altered by fusing its minimal F-box to a synthetic anti-GFP nanobody (VHH-GFP4; Rothbauer et al., 2008) replacing the C-terminal WD40 repeat domain (Caussinus et al., 2011). Nanobodies are small proteins (10–15 kDa) that resemble the variable fragment (VHH) of homodimeric heavy-chain antibodies, derived from camelids (Harmsen and De Haard, 2007). They are highly stable, can be easily expressed in mammalian cells, and bind strongly to their specific antigens (Beghein and Gettemans, 2017). Upon inducible expression of the F-box-nanobody fusion protein, the GFP-tagged POI is bound, endogenous SKP1, CUL1, and RBX1 are recruited, and ubiquitylation and degradation are initiated. The deGradFP system degrades GFP-tagged proteins in *Drosophila* with a half-life of 2 h and is also functional for several GFP derivates like Venus, yellow fluorescent protein (YFP), and enhanced YFP (Caussinus and Affolter, 2016; Caussinus et al., 2011). It was successfully adapted to other model organisms, such as *Danio rerio* and *Trypanosoma brucei* (Ishii and Akiyoshi, 2022; Yamaguchi et al., 2019), but so far failed to degrade nuclear proteins in mammalian cells (Daniel et al., 2018; Ludwicki et al., 2019; Shin et al., 2015).

In the AID systems, the plant hormone auxin (indole-3-acetic acid, IAA or AUX) induces target protein degradation via the SCFs osTIR1 from *Oriza sativa* (Nishimura et al., 2009), or aaAFB2 from *Arabidopsis thaliana* (Li et al., 2019). Upon AUX binding, osTIR1 undergoes a structural change allowing stable interactions with AID sequences in target proteins, which are subsequently ubiquitylated and degraded (Holland et al., 2012; Nishimura et al., 2009; Trost et al., 2016). AID sequences can be fused directly to the target proteins or to nanobodies, which can be inducibly expressed and bind to the target protein. Upon addition of AUX, degradation occurs within 20–60 min, and it is stopped upon AUX withdrawal (Daniel et al., 2018; Holland et al., 2012; Li et al., 2019; Nishimura et al., 2009). Despite these advantages, AID systems require the constitutive expression of large foreign FBPs, which is often toxic for cells, and they are leaky showing basal degradation without AUX induction.

An alternative degradation system that functions independently of the SCF complex is called Trim-Away and is based on the E3-Ub ligase tripartite motif-containing protein 21 (TRIM21), which confers target specificity and E3-Ub ligase activity (Clift et al., 2017). TRIM21 recognizes and binds the Fc region of antigen-bound antibodies (IgG) and mediates their ubiquitylation and degradation (James et al., 2007). This system enables a rapid and efficient degradation of cytoplasmic proteins, with half-lives of 10–30 min. However, target-specific antibodies need to be delivered into cells by electroporation which, due to their large size, cannot diffuse through the nuclear pore and thus cannot be used to deplete nuclear proteins (Clift et al., 2017).

Here, we present hGRAD, short for human protein degradation system, which is optimized to induce the rapid and efficient degradation of GFP-tagged nuclear RBPs from various subnuclear compartments. hGRAD allowed us to identify a novel function of SRSF5 in promoting transcript stability and to define direct target genes and compensatory mechanisms.

## Results

### Development of the hGRAD system

To study the functions of nuclear RBPs, we aimed to develop a rapid degradation system that is easy to implement, requires minimal engineering of host cells, and efficiently degrades RBPs that localize to nuclear condensates. Ideally, it should function in different species, cell types, and subcellular compartments, and allow to follow the kinetics of degradation in real time. We decided to target GFP-tagged proteins since many groups use endogenous GFP tagging by CRISPR/Cas9 to study the functions of condensate RBPs, and many tagged cell lines are commercially available.

We first designed a "one-fits-all" plasmid expression system, which allows the inducible expression of two genes in a doxycycline (DOX)-dependent manner (Fig. 1 A). This vector, named pTRE-BI, expresses a Tet-regulator (Tet-on 3G) from the *EF-1α* promoter, which drives robust constitutive expression independently of mammalian cell type and species. It also contains a bi-directional Tet-inducible promoter (TRE3G BI) with two flanking multiple cloning sites (MCS) and a puromycin resistance gene for genomic integration and selection (Fig. 1 A).

Next, we modified the deGradFP system (Caussinus et al., 2011) since previous studies indicated that deGradFP is inefficient in degrading nuclear proteins in mammalian cells (Daniel et al., 2018). In humans, >70 different FBPs have been reported that mediate target selection and specificity (Frescas and Pagano, 2008; Jin et al., 2004; Zheng et al., 2002). We identified the human protein FBXW11 as the closest ortholog to *Drosophila* Slmb (Fig. 1 B and Fig. S1 A) and fused its minimal F-box domain to an anti-GFP nanobody (VHH-GFP4). The F-box-nanobody fusion was inserted on one side of the bi-directional TREG3 BI promoter and mCherry as induction control on the other side. The Tet-regulator activates the transcription of both genes upon DOX addition. We named our "one-fits-all" plasmid system hGRAD for human protein degradation system (Fig. 1 C). Random genomic integration of this vector should allow the inducible degradation of any GFP-tagged protein through binding of the anti-GFP nanobody and recruitment of the cellular SCF

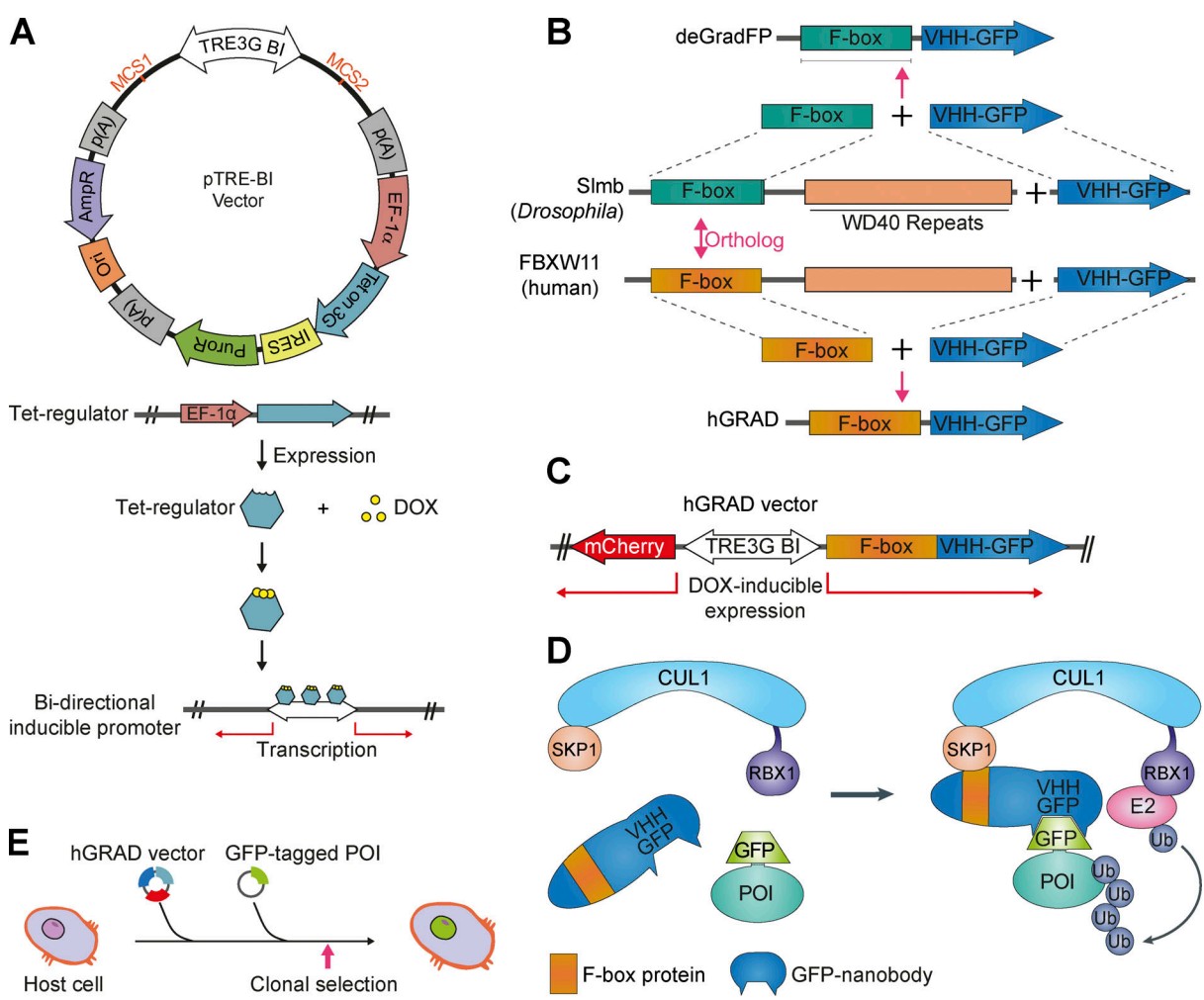

Figure 1.  **Development of the hGRAD system. (A)** Scheme of the pTRE-BI one-fits-all backbone vector. p(A), polyadenylation sites; *EF-1α*, promoter; Tet on 3G, gene encoding Tet-regulator; IRES, internal ribosomal entry site; PuroR, puromycin resistance gene; Ori, origin of replication; AmpR, ampicillin resistance gene. **(B)** The hGRAD system is based on the deGradFP system (Caussinus et al., 2011) and was optimized for mammalian protein degradation by exchanging the *Drosophila* Slmb F-box domain with the human FBXW11 F-box domain. **(C)** Scheme of the pTRE-BI hGRAD vector. **(D)** Overview of the hGRAD system harnessing the cellular SCF complex for ubiquitylation and degradation of the GFP-tagged POI. Model modified from Yamaguchi et al. (2019). **(E)** Random genomic integration of the pTRE-BI-hGRAD plasmid allows for inducible degradation of any GFP-tagged protein in any mammalian host cell type.

complex, which mediates ubiquitylation and proteasomal degradation (Fig. 1, D and E).

### hGRAD degrades nuclear RBPs more efficiently than other degradation systems

To be able to compare the performance of hGRAD in degrading mammalian GFP-tagged proteins with other rapid degradation systems, we modified the target specificity of osTIR1, aaAFB2 (AID systems), and TRIM21 (Trim-Away system) with anti-GFP nanobodies and cloned all necessary protein-coding sequences in the pTRE-BI vector (Fig. 2 A): genes encoding osTIR1 and aaAFB2 were fused to mCherry and to a weak nuclear localization signal (NLS) at their C-terminus that was taken from Li et al. (2019) to ensure optimal degradation of cytoplasmic and nuclear proteins. The gene fusions were then inserted on one side of the bi-directional promoter. On the other side, a sequence encoding a GFP nanobody fused to the minimized AID (mAID) sequence was inserted for the osTIR1 system (Daniel et al., 2018),

while for the aaAFB2 system, an optimized AID sequence (miniIAA7) was used (Li et al., 2019). For the Trim-Away system, the *TRIM21* gene was also fused to mCherry and a weak NLS, and the anti-GFP nanobody was fused to the Fc region of human IgG1 (hIgG-Fc2; Clift et al., 2017; Fig. 2 A).

We generated four master cell lines by integrating the different pTRE-BI constructs into the genome of wild-type (WT) HeLa cells and selecting single clones. All pTRE-BI–based induction systems showed a strong DOX-induced expression of the desired proteins and no expression in the uninduced state (Fig. 2 B). Despite their large molecular weights (>100 kDa), all mCherry fusion proteins localized to both the cytoplasm and the nucleus (Fig. S1 B), demonstrating that the weak NLS is functional. Thus, targeted protein degradation should occur in both subcellular compartments (Franić et al., 2021).

We next performed proliferation assays to evaluate potential adverse effects of the DOX treatment. The four master cell lines were grown in the presence or absence of DOX (1 µg/ml; 72 h),

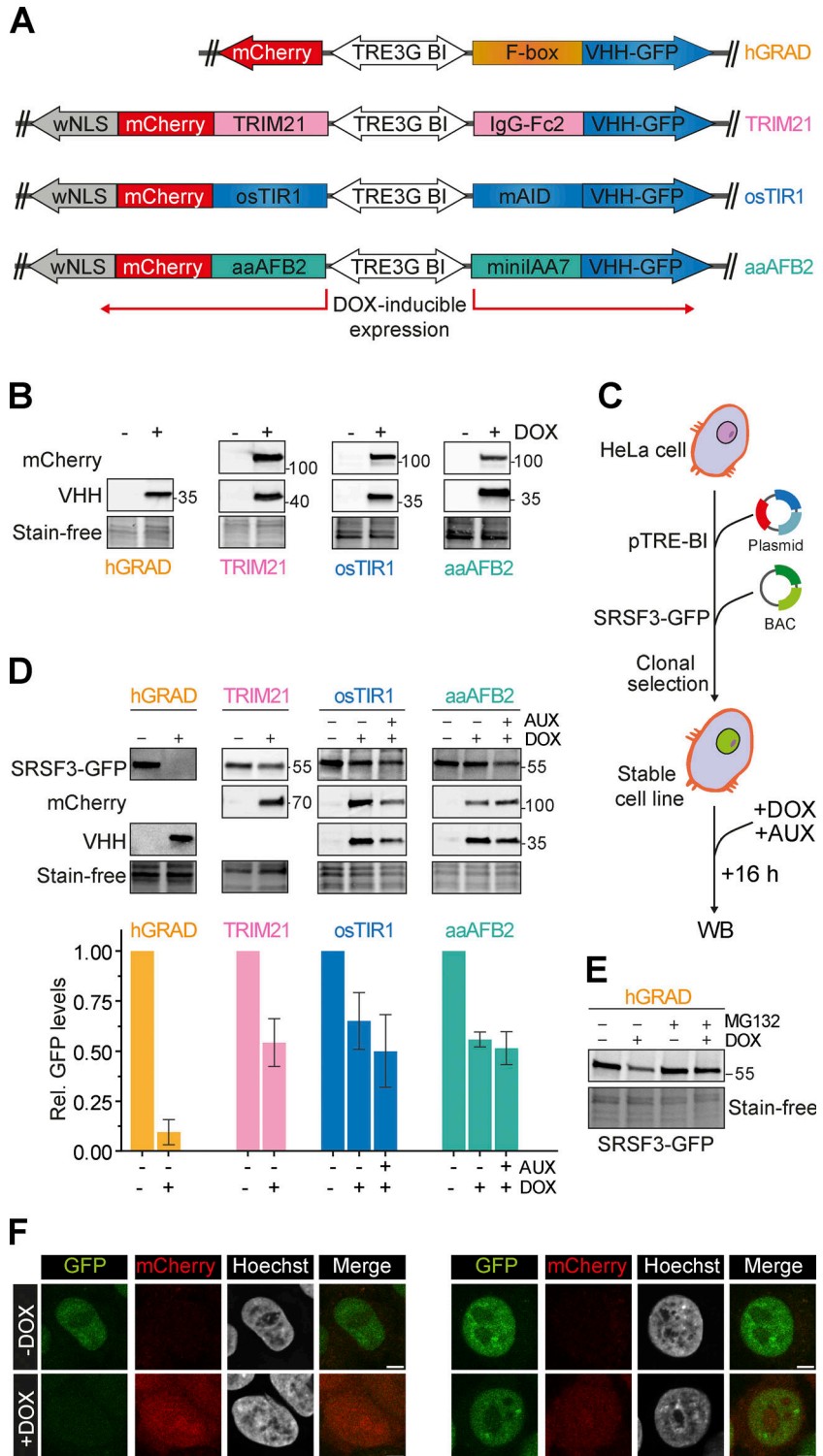

**A**

**B**

**C**

**D**

**E**

**F**

Figure 2. **hGRAD degrades the nuclear RBP SRSF3 more efficiently than other degradation systems in HeLa cells. (A)** Schemes of the pTRE-BI one-fits-all vectors for expression of the hGRAD, TRIM21, osTIR1, and aaAFB2 systems. **(B)** Induction in HeLa cells (16 h, 1 μg/ml DOX, 0.5 mM AUX). TRIM21-, osTIR1-, and aaAFB2-mCherry fusion proteins were detected by an anti-mCherry antibody. The different nanobody fusions (FBXW11-, IgG-Fc2-, mAID-, and miniIAA7-VHH-GFP) were detected by an anti-camelid VHH antibody. Stain-free gels were used to detect total protein and to control for equal loading. **(C)** Experimental scheme and comparison of the degradation efficiencies of hGRAD, TRIM21, osTIR1, or aaAFB2 systems after 16 h DOX and AUX induction. WB, western blot. **(D)** SRSF3-GFP, mCherry, and the nanobody fusions were detected with anti-GFP, anti-mCherry, and anti-camelid VHH antibodies, respectively. Shown are a representative western blot and the quantification of *n* = 3 independent replicates. Error bars show standard deviation of the mean (SD). **(E and F)** (E) Western blot and (F) micrographs show that SRSF3-GFP degradation is blocked upon proteasome inhibition with MG132 (10 μM 4 h). Scale bars = 5 μm. Source data are available for this figure: SourceData F2.

and doubling times ($t_d$) were calculated. The osTIR1, aaAFB2, TRIM21, and hGRAD cell lines showed similar growth rates compared with WT HeLa cells in the absence of DOX ($t_d$ = 17.23 h), suggesting that the integrated pTRE-BI plasmid itself does not negatively affect cell growth (Fig. S1 C). However, upon DOX addition, the osTIR1 ($t_d$ = 19.3 h), aaAFB2 ($t_d$ = 20.84 h), and TRIM21 ($t_d$ = 18.41 h) cell lines showed reduced growth rates,

indicating potential cytotoxic effects from the enzymatic activity or expression of the heterologous proteins. In contrast, the hGRAD cells showed similar doubling times ($t_d$ = 17.24 h) to WT cells, indicating that expression of the hGRAD components does not affect cell growth (Fig. S1 C).

To compare the efficiency in targeting and degrading GFP-tagged SR proteins, we inserted a bacterial artificial chromosome

(BAC) harboring the *SRSF3* gene fused to a GFP tag into the genome of all four master cell lines (Figs. 2 C and S1 D). The BAC gene harbors the complete *SRSF3* gene, including the endogenous *SRSF3* promoter, untranslated regions (UTRs), exons, and introns, so that *SRSF3-GFP* is expressed, processed, and regulated similarly to the endogenous gene (Müller-McNicoll et al., 2016). All four cell lines showed similar SRSF3-GFP levels and the expected localization to the nucleoplasm and nuclear speckles (Fig. S1 E).

After DOX treatment (16 h), all proteins required for rapid degradation were strongly expressed (Fig. 2 D). The TRIM21, osTIR1, and aaAFB2 systems reduced SRSF3-GFP levels but only by about 50% in the presence of AUX (16 h). Moreover, in the osTIR1 and aaAFB2 systems, SRSF3-GFP levels were already reduced in the presence of DOX without AUX, indicating that the AID system is leaky, in line with previous findings (Verma et al., 2020). In contrast, the hGRAD system showed an almost complete depletion of SRSF3-GFP (Fig. 2 D). Depletion was blocked when the proteasome was inhibited by MG132 treatment (Fig. 2, E and F), confirming that degradation occurs via the ubiquitin-proteasome system (UPS).

Our data demonstrate that the one-fits-all pTRE-BI vector backbone allows a strong and tightly controlled inducible expression of the factors required for degradation of both nuclear and cytoplasmic proteins. Compared with the osTIR1, aaAFB2, and TRIM21 systems, hGRAD showed the most efficient depletion of a nuclear RBP and the mildest cytotoxic effects.

## hGRAD efficiently degrades RBPs that localize to nuclear condensates

We next evaluated the capacity of hGRAD to degrade GFP-tagged proteins in different subcellular compartments. For this, we stably integrated BACs expressing the eukaryotic translation initiation factor 4E (EIF4E-GFP) and the non-RBP serine/arginine-rich protein-specific kinase 1 (SRPK1-GFP) as cytoplasmic marker proteins, SRSF5-GFP as marker for nuclear speckles, and non-POU domain-containing octamer-binding protein (NONO-GFP) as a marker for paraspeckles into the HeLa hGRAD cell line (Fig. 3 A). The correct subcellular localization of the GFP-tagged proteins was confirmed by confocal microscopy (Fig. 3 D and Fig. S2 A). Western blot experiments revealed that the hGRAD-mediated degradation worked most efficiently for proteins localizing to nuclear speckles (SRSF3 and SRSF5) or paraspeckles (NONO; >90%), while cytoplasmic proteins were degraded slightly less efficiently (EIF4E and SRPK1; Fig. 3 B). This was confirmed by confocal microscopy (Fig. S2 A).

To separate direct from indirect effects, a fast depletion of a protein is most desirable. To evaluate the kinetics of hGRAD-mediated degradation, we performed a time-course for SRSF3, SRSF5, NONO, and EIF4E (0, 2, 4, 6, and 8 h DOX; Fig. 3 C). Proteins were already reduced 2 h after induction, concomitant with expression of the anti-GFP nanobody. The depletion efficiency differed among the GFP-tagged proteins. Nuclear SRSF3-, SRSF5-, and NONO-GFP were degraded rapidly, with half-lives of 2.0 and 2.5 h and almost 90% depletion after 6 and 8 h, respectively. Cytoplasmic EIF4E-GFP was degraded slightly slower, with

a half-life of 3.3 h and 75% depletion after 8 h (Fig. 3 C and Fig. S2 B).

To monitor degradation in individual cells over time, we combined rapid degradation with confocal live-cell imaging. HeLa hGRAD cell lines expressing SRSF3-GFP or SRSF5-GFP were imaged in the presence or absence of DOX for 12 h. Relative GFP intensity was quantified in individual cells and normalized to the first uninduced timepoint (T0). Similar to the western blot quantifications, SRSF3-GFP showed a half-life of 2.2 h and SRSF5-GFP of 2.5 h (Fig. 3 D). Both proteins reached their minimum levels (below 10% compared to T0) after 4.0 h and 4.5 h of induction, respectively (Fig. 3 E). This indicates that hGRAD degradation of nuclear proteins in individual cells has similar kinetics to bulk cell measurements. Thus, hGRAD quickly and efficiently clears proteins from nuclear condensates within 3 h.

## hGRAD works efficiently in different mammalian cell types and species

We next evaluated the capacity of hGRAD to degrade GFP-tagged proteins in pluripotent mouse P19 cells. For this, we generated a P19 hGRAD master cell line and integrated BACs expressing SRSF3-, EIF4E-, SRPK1-, and NONO-GFP into the genome (Fig. S2 B). Similar to HeLa cells, the anti-GFP nanobody fusion was well expressed upon DOX induction (1 µg/ml) and undetectable in the absence of DOX in P19 cells, indicating strong inducibility and no leaky expression (Fig. 4 A). Proliferation assays revealed that neither integration of the pTRE-BI-hGRAD vector itself (uninduced) nor DOX induction showed any significant changes in growth rates compared with WT P19 cells with the characteristic doubling time of 12 h (Fig. 4 B).

Endpoint degradation assays and GFP fluorescence microscopy confirmed the efficient depletion of all tested proteins in P19 cells (Fig. 4 C and Fig. S2 A). SRSF3- and NONO-GFP showed a half-life of 2.9 h after induction and EIF4E-GFP of 4.3 h (Fig. 4 D and Fig. S2 B). Thus, our data demonstrate that the hGRAD system allows efficient degradation of GFP-tagged proteins in both human HeLa and mouse P19 cells.

## hGRAD works as a rapid knockdown tool for endogenously GFP-tagged nuclear RBPs

We next tested the capacity of hGRAD in degrading endogenously GFP-tagged proteins (endo-GFP) as a rapid knockdown tool. For this, we inserted a GFP tag at the C-termini of *SRSF3*, *SRSF5*, and the nuclear speckle marker *SRRM2* in the HeLa and P19 hGRAD master cell lines using CRISPR/Cas9 (Fig. 5 A). Successful GFP tagging of single clones was validated by PCR, Sanger sequencing, and western blot (Fig. S3, A–D). Confocal microscopy revealed the correct nuclear localization of SRSF3-endo-GFP and SRSF5-endo-GFP, with their typical distribution in the nucleoplasm and enrichment within nuclear speckles, as well as of SRRM2-endo-GFP, which exclusively localized to nuclear speckles (Fig. S3 E). We concluded that all endogenously tagged proteins were correctly folded and fully functional.

Endpoint degradation assays revealed that SRSF3-endo-GFP and SRSF5-endo-GFP were efficiently depleted in HeLa cells (by >90%; Fig. 5 B), similar to the system where they were expressed

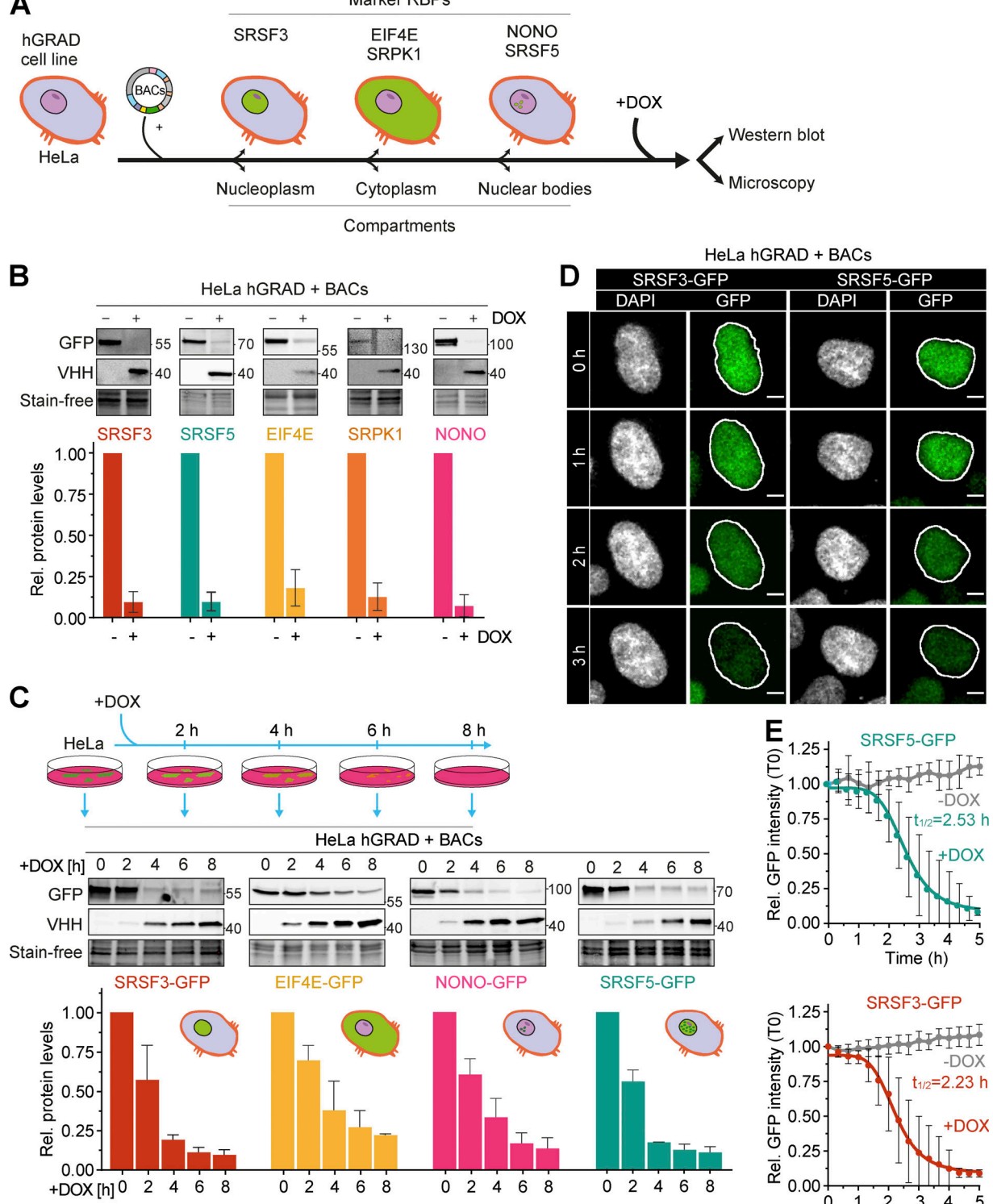

Figure 3. **hGRAD works efficiently for RBPs that localize to nuclear condensates. (A)** Experimental scheme to test the hGRAD degradation for human proteins localizing to different subcellular compartments. **(B)** Comparison of the degradation efficiency of GFP-tagged RBPs in HeLa cells 16 h after DOX induction (1 µg/ml). **(C)** Degradation time courses show that hGRAD works most efficiently for nuclear RBPs. **(B and C)** Shown are the quantifications of $n = 3$ independent western blot experiments and one representative blot relative to the −DOX or 0 h timepoint. GFP-tagged proteins and the hGRAD nanobody fusion were detected with anti-GFP and anti-camelid VHH antibody, respectively. **(D)** Live-cell imaging to follow the hGRAD degradation kinetics. **(E)** Quantification of half-lives ($t_{1/2}$) of SRSF3-GFP and SRSF5-GFP in HeLa cells upon DOX induction. Scale bars = 5 µm. Error bars, SD. Source data are available for this figure: SourceData F3.

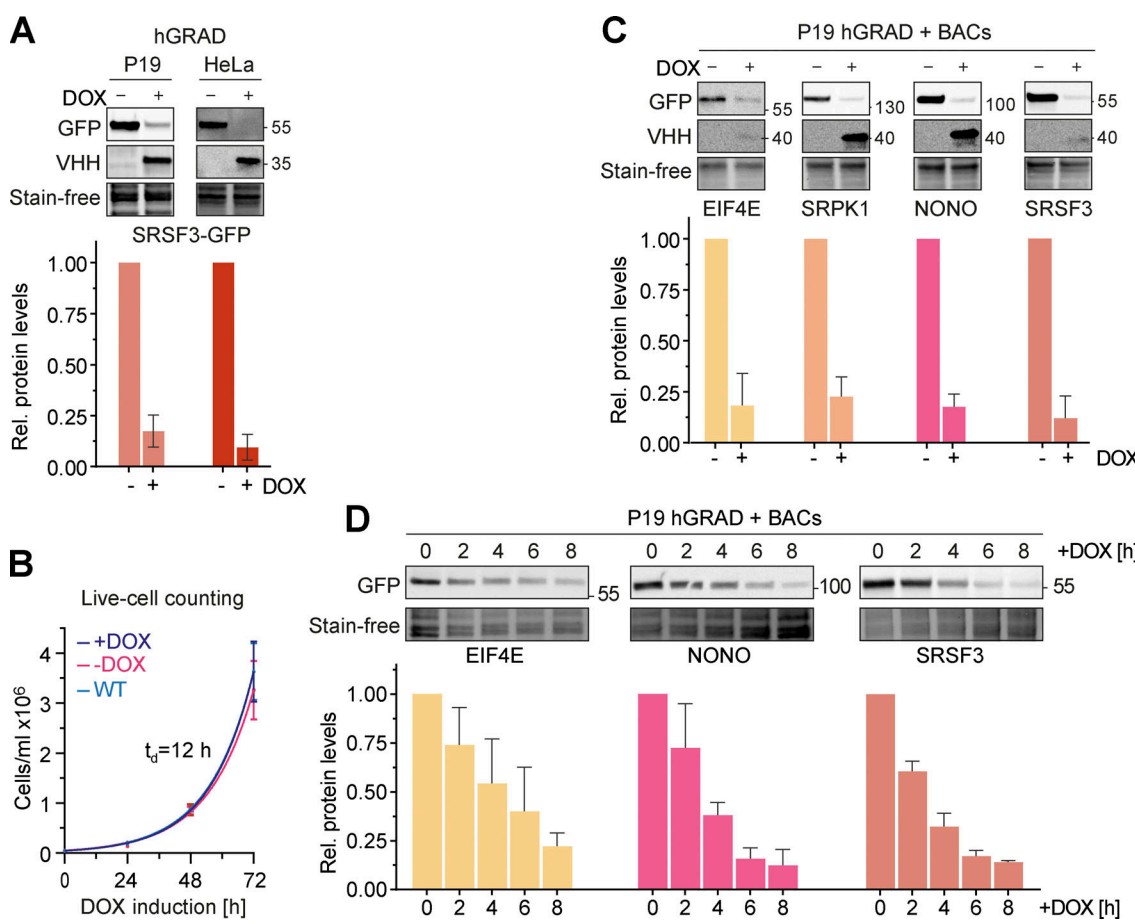

Figure 4. **hGRAD works efficiently in different mammalian cell types and species. (A)** Comparison of the degradation efficiency of GFP-tagged SRSF3 in HeLa and P19 cells 16 h after induction with DOX (1 µg/ml). **(B)** P19 hGRAD proliferation assay. Growth curves and $t_d$ were evaluated by exponential growth equation fit ($Y=Y_0*exp(k*X)$). P19 WT (light blue), P19 hGRAD –DOX (pink), and P19 hGRAD +DOX (dark blue). Shown are mean and SD (error bars) from $n = 3$ independent experiments. **(C)** Comparison of the degradation efficiency of GFP-tagged proteins in mouse P19 cells 16 h after DOX induction. **(D)** Degradation time courses show that hGRAD works efficiently in mouse P19 cells. **(A–D)** Shown are one representative western blot and the quantifications of $n = 3$ independent experiments (relative to the –DOX or 0 h timepoint). GFP-tagged proteins and the hGRAD nanobody fusion were detected with anti-GFP and anti-camelid antibody, respectively. Error bars = SD. Source data are available for this figure: SourceData F4.

from BACs (see above). Time-course experiments indicated a half-life of 2.2 h for SRSF3-endo-GFP (Fig. 5 C), reaching its minimum after 4 h, similar to SRSF3-GFP from the BAC ($t_{1/2}$ = 2.0 h, Fig. 3 C). SRRM2-endo-GFP showed a half-life of 2.2 h and reached its minimum after 6 h, demonstrating that hGRAD is capable of degrading nuclear proteins that form the core of nuclear speckles (Fig. 5 D). SRSF5-endo-GFP showed a half-life of 2.1 h and reached its minimum levels after 4 h (Fig. 5 C). For comparison, the BAC-expressed SRSF5-GFP showed a half-life of 2.5 h, suggesting that SRSF5-endo-GFP and SRSF5-GFP were degraded with similar efficiencies (Fig. 3 C and Fig. 5 C).

Degradation of SRSF5-endo-GFP was blocked when cells were treated with the proteasome inhibitor MG132 (Fig. S3 F). In P19 cells, degradation kinetics of SRSF3-endo-GFP and SRSF5-endo-GFP were similar to those in HeLa cells (Fig. S4, A and B); both proteins were depleted by 80–90% after 6 h. Live-cell imaging confirmed that SRSF3-endo-GFP and SRSF5-endo-GFP were depleted slightly faster than their counterparts expressed from BACs, with half-lives of 2.13 and 2.15 h, respectively (Fig. 5, E and F).

Interestingly, hGRAD depletion of SRSF5-endo-GFP worked with similar efficiency, independently of the glucose concentration (Fig. S4 D), although a previous study proposed that SRSF5 is much less stable under low glucose conditions (Chen et al., 2018). We also did not observe any differences in SRSF5-endo-GFP protein half-lives in high and low glucose conditions (Fig. S4 C). Altogether, our data show that hGRAD efficiently degrades endogenously GFP-tagged SRSF3, SRSF5, and SRRM2, the former shown in HeLa and P19 cells. This underlines that the simple one-step integration of the hGRAD plasmid into any cell line that expresses endogenously GFP-tagged proteins allows their rapid and inducible knockdown.

## Combining hGRAD with nascent RNA sequencing (Nascent-seq) reveals dynamic changes in transcript levels

Conventional knockdown of SRSF5 using siRNAs required 72 h to obtain a 75% reduction in the protein level (Fig. S5 A). These long depletion times render it difficult to distinguish direct from indirect targets or identify new functions of SRSF5 due to compensatory mechanisms. To improve our understanding of

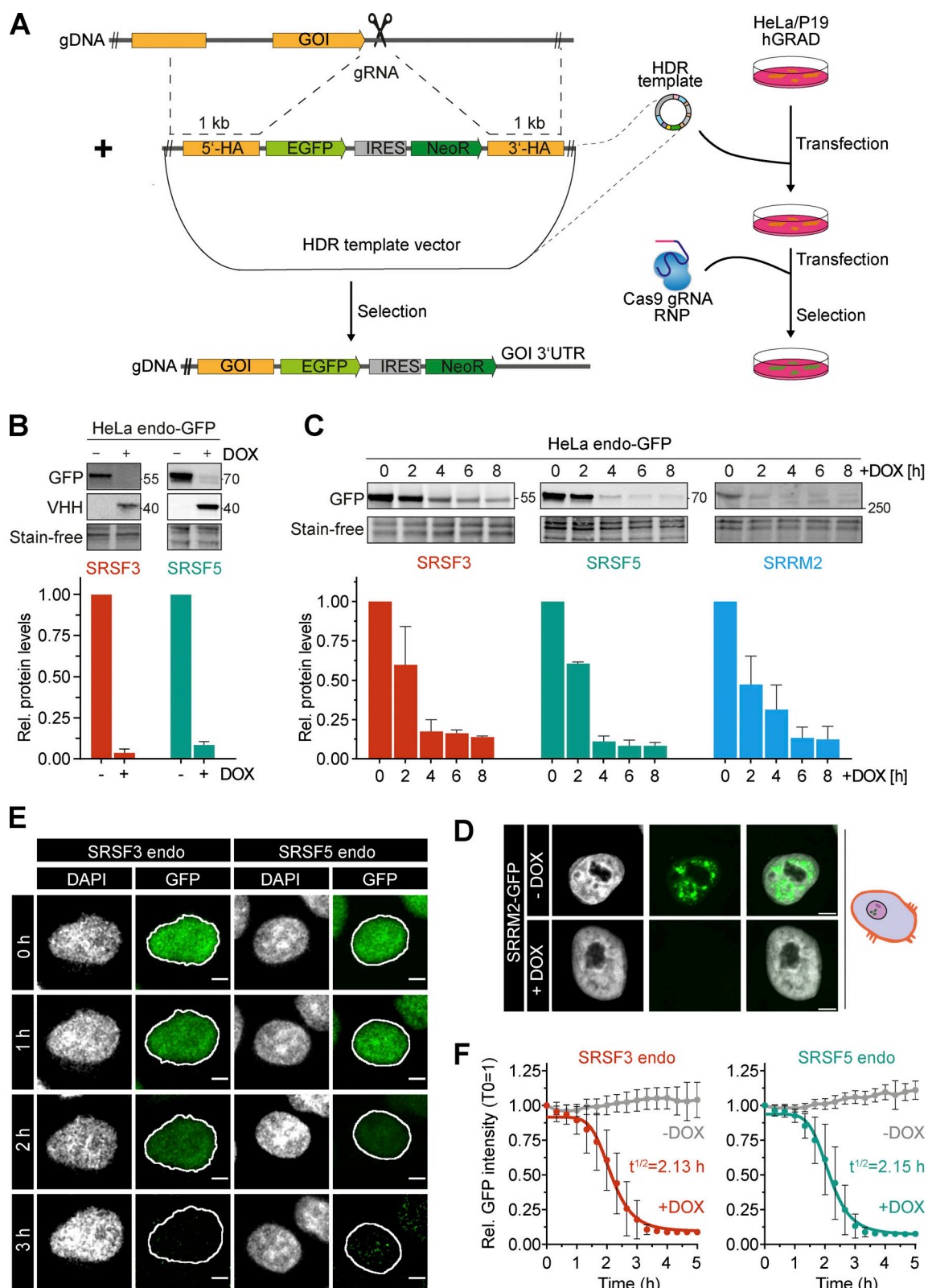

Figure 5.    **hGRAD efficiently degrades endogenously GFP-tagged nuclear RBPs in HeLa cells, allowing their rapid knockdown. (A)** Experimental scheme for the endo-GFP tagging using CRISPR/Cas9 in HeLa and P19 hGRAD cells. gDNA, genomic DNA; gRNA, guide RNA; GOI, gene of interest; HDR, homology-directed repair; NeoR, Neomycin resistance cassette; 5'- and 3'-HA, homology arms. **(B)** Comparison of the degradation efficiency of SRSF3- and SRSF5-endo-GFP in HeLa cells after 16 h of induction by DOX. **(C)** Degradation time courses show that hGRAD works efficiently for SRSF3-, SRSF5-, and SRRM2-endo-GFP. **(B and C)** Shown are a representative western blot and the quantification of $n$ = 3 independent experiments. Error bars = SD. GFP-tagged proteins and hGRAD nanobody fusion proteins were detected with anti-GFP and anti-camelid antibody, respectively. **(D)** Degradation of SRRM2-GFP from nuclear speckles. Scale bars = 5 µm. **(E)** Live-cell imaging reveals hGRAD degradation kinetics. Scale bars = 5 µm. **(F)** Quantification of the half-lives ($t_{1/2}$) of SRSF3-endo-GFP and SRSF5-endo-GFP in HeLa cells. Source data are available for this figure: SourceData F5.

this understudied splicing factor, we combined rapid depletion with Nascent-seq (Rädle et al., 2013; Schwalb et al., 2016). Nascent-seq captures only changes that occur in newly transcribed RNAs and ignores the large fraction of pre-existing nuclear and cytoplasmic RNAs that would mask these changes. Especially at early timepoints, only a small number of differentially expressed or alternatively spliced targets are expected.

Homozygously tagged HeLa SRSF5-endo-GFP cells (Fig. S3 D) were induced with DOX and at different timepoints (2, 8, and 16 h), 400 μM 4-thio-uridine (4sU) was added for 1 h, which is incorporated into newly transcribed RNAs (Fig. 6 A). Uninduced cells were used as controls (T0), and SRSF5 depletion was controlled by western blot (Fig. S5 B). After total RNA extraction, nascent transcripts with incorporated 4sU were biotinylated in vitro, purified using streptavidin-coated beads, and converted into a strand-specific cDNA library (Fig. 6 A and Fig. S5 C). 40% of uniquely mapped reads corresponded to intronic regions, in line with enrichment of nascent, incompletely spliced pre-mRNAs. Surprisingly, differential expression analysis using DESeq2 (Love et al., 2014) uncovered very rapid and dynamic changes in gene expression after SRSF5 depletion (Fig. 6, B and C; and Table S1). At T2, already 436 genes were differentially expressed (adjusted P value <0.05; Fig. 6 B). Of those, 127 were exclusively regulated at T2, whereas 178 differentially expressed genes (DEGs) changed over the entire time course. Interestingly, at T2, 87% of all DEGs were downregulated (Fig. S5 D). Among them were 106 (28%) long non-coding RNAs (lncRNAs; Fig. 6 C). 50% of the downregulated lncRNAs remained reduced until T16, while 40% were no longer regulated, and 10% were even upregulated at T16 (Fig. 6 C). T8 showed the highest number of DEGs (2,531) with 1,766 exclusively regulated genes at this timepoint, while at T16, the number of DEGs was reduced again to 1,079, with 298 exclusively regulated genes (Fig. 6 B). This suggests that at longer SRSF5 depletion times, changes in gene expression are attenuated or overshot due to compensatory mechanisms. In line with this, we identified very few DEGs upon siRNA-mediated SRSF5 knockdown (48 h) in P19 cells (Müller-McNicoll et al., 2016).

**Downregulated transcripts after SRSF5 depletion are direct targets of SRSF5**

To test whether downregulated transcripts are bound by SRSF5, we performed individual-nucleotide resolution UV crosslinking and immunoprecipitation (iCLIP2; Buchbender et al., 2020) with HeLa hGRAD SRSF5-endo-GFP cells (five replicates; Fig. S5 E). We identified 415,245 reproducible SRSF5 binding sites in 8,280 protein-coding genes, mainly in coding regions and 5′UTRs, and 708 lncRNAs (Fig. S5, F and G; and Table S1). Interestingly, 58% of the transcripts downregulated at T16 contained SRSF5 binding sites compared with 15% in unchanged and 8% in upregulated transcripts. A similar trend was observed for the other timepoints (Fig. S5 H), suggesting that those are direct targets of SRSF5. Consistently, highly bound transcripts tended to be more downregulated upon SRSF5 depletion (T8 and T16) than unbound or lowly bound transcripts (Fig. 6 D). For example, the lincRNAs LINC01895, BASP1-AS, ENSG00000231412, and ENSG00000259621 as well as the mRNA encoding for SRSF2 were

extensively bound by SRSF5 and showed decreased levels at all three timepoints (Fig. 6 E and Fig. S5 I). We confirmed the decrease of ENSG00000231412 and ENSG00000259621 transcripts after acute SRSF5 depletion by quantitative RT-PCR (RT-qPCR), which was not seen after similar induction times of the hGRAD master cell line (Fig. 6 F), indicating that the observed effects are direct and not due to induction of the hGRAD system.

Unexpectedly, one mRNA that was strongly bound by SRSF5 and downregulated at T8 and T16 was the SRSF5 transcript itself. We expected its mRNA levels to rise upon depletion of the SRSF5 protein since SRSF5 autoregulates its own mRNA levels via splicing of a poison cassette exon (PCE), similar to other SR proteins (Müller-McNicoll et al., 2019). RT-qPCR confirmed that SRSF5 transcript levels decreased to 50% and remained low even at 32 h of SRSF5 depletion (Fig. 6 G). SRSF5 bound most strongly in a region surrounding the SRSF5 PCE, suggesting that SRSF5 binding normally regulates PCE inclusion (Fig. 6 H). In line with this, quantification of PCE junctions revealed a decrease in PCE inclusion at T8 and T16 from 10% to 4% (Fig. S5, J and K). This means that the levels of SRSF5-PCE isoforms decreased, while those of translatable SRSF5 isoforms increased at T16, which might be a way to counteract degradation of SRSF5.

Apart from SRSF2 and SRSF5, no other SR protein–encoding mRNAs changed upon SRSF5 depletion. However, we observed a steady increase in PCE inclusion from T0 to T16 in SRSF3 (from 57.7% to 64.9%), SRSF6 (from 42.8% to 54.9%), and SRSF7 (from 71.1% to 81%; Fig. S5 J). Fewer translatable isoforms of these SR proteins may contribute to the observed compensatory effects at longer depletion times. Thus, our data indicate that auto- and crossregulation via PCE inclusion also operate when SR protein levels are too low.

What causes the early decrease in the levels of SRSF5 mRNA and other transcripts? SRSF5 could promote their transcription, splicing, 3′ end processing, and/or stability. Since the sudden decrease in SRSF5, ENSG00000231412, and ENSG00000259621 levels was also detectable by qPCR with total RNA (Fig. 6, F and G), which is dominated by "old" transcripts, reduced transcription could be ruled out. To test whether SRSF5 depletion impacts splicing, we analyzed the Nascent-seq data using MAJIQ (Jha et al., 2017). We observed only a few splicing changes (Fig. S5 K), indicating that SRSF5 might not be a major splicing factor and may fulfill other functions. The most affected type of splicing events was intron retention (IR), indicating slower splicing, whereby many events were transiently regulated at T8 (Fig. S5, K–M). IR was confirmed by analysis with IRFinder (Lorenzi et al., 2021), which identified 181 significant IR events at timepoint T2, 717 events at T8, and 142 events at T16 (Table S1). Only 1.1% of the downregulated transcripts showed IR at T2 (7.1% at T8, 1.4% at T16), and there was no evidence of premature polyadenylation. Together, this suggests that alternative splicing and IR are not the main cause of decreased transcript levels.

We thus speculated that SRSF5-PCE isoforms and other non-coding transcripts might decrease in abundance after acute SRSF5 depletion because SRSF5 normally binds them and protects them from nuclear decay machineries, e.g., by hiding them in nuclear speckles. To test this, we combined SRSF5 hGRAD depletion with RNA fluorescence in situ hybridization (FISH).

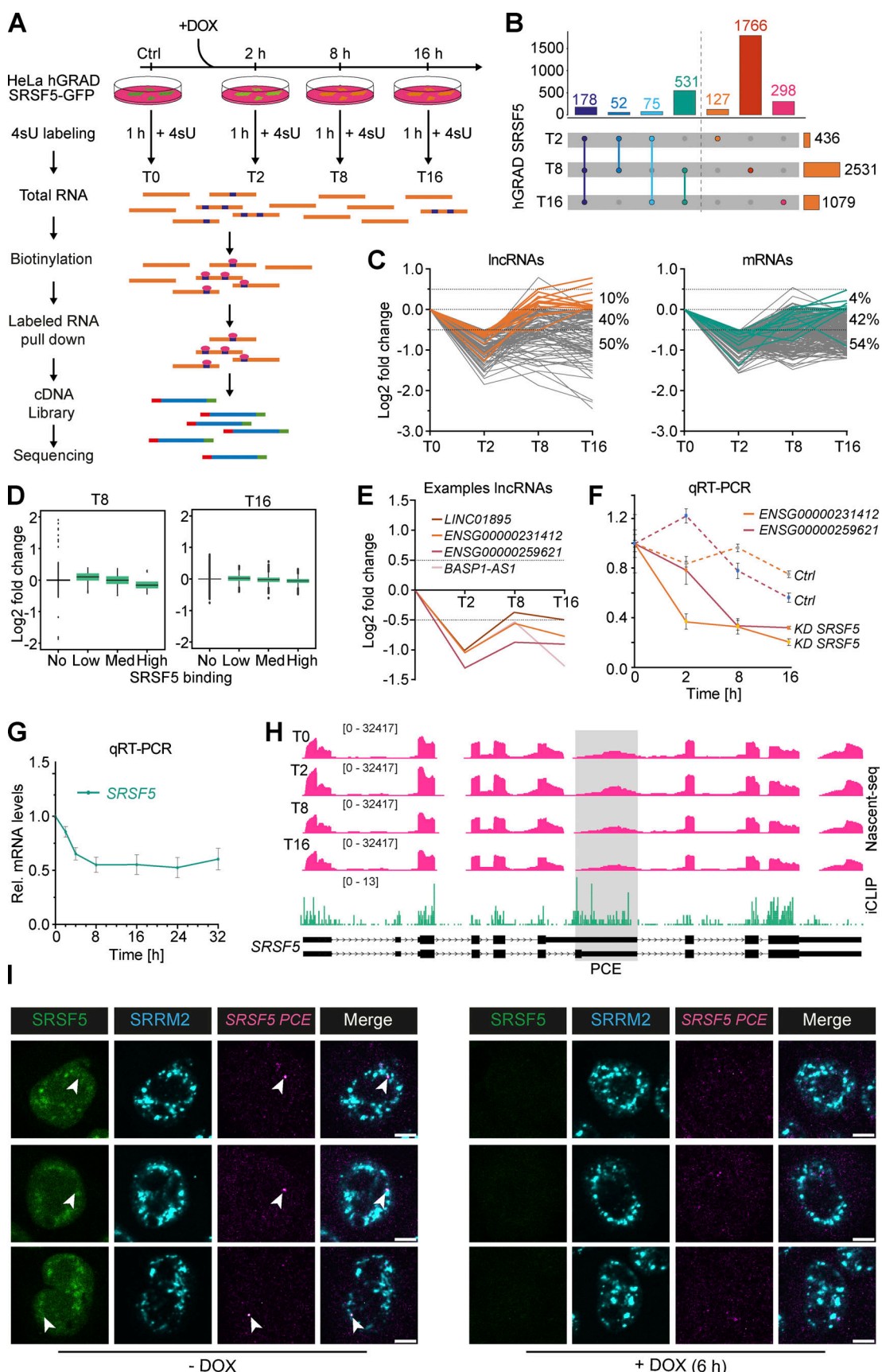

Figure 6. **Combining hGRAD with Nascent-seq and iCLIP2 allows the identification of direct SRSF5 targets and reveals a function of SRSF5 in transcript stabilization. (A)** Scheme of the workflow combining a time course of hGRAD degradation with Nascent-seq. Cells were first induced with DOX

(1 µg/ml) for the indicated times and then treated additionally with 4sU (400 µM) for 1 h to label all new transcripts. Subsequently, total RNA was extracted, biotinylated, purified, and converted into a cDNA library for deep sequencing. **(B)** Upset plot of differentially expressed transcripts. P value >0.05. **(C)** Rapid depletion of SRSF5 by hGRAD leads to dynamic changes in the abundance of mRNAs (right) and lncRNAs (left). All RNAs downregulated at T2 are shown in gray. Transcripts that are compensated at T16 are highlighted in orange (lncRNAs) or green (mRNAs). **(D)** Log$_2$ fold change distribution of transcripts at T8 and T16 separated into non, low, medium, and highly bound transcripts based on the normalized number of binding sites. **(E)** Examples of the dynamics of downregulated lncRNAs. **(F)** RT-qPCR to show that lncRNA downregulation is specific to SRSF5 depletion. **(G)** RT-qPCR to show that *SRSF5* mRNA decreases until 8 h and remains low until 32 h after induction. Graph shows mean and SD of *n* = 3 independent experiments. **(H)** Genome browser view shows Nascent-seq and SRSF5 iCLIP2 crosslinks on the *SRSF5* gene. **(I)** Left: Representative micrographs show the subcellular localization of SRSF5-endo-GFP and *SRSF5-PCE* isoforms in HeLa cells before DOX induction. Right: Representative micrographs show the degradation of SRSF5-endo-GFP and the reduction of *SRSF5-PCE* isoforms in HeLa cells after 6 h induction by DOX (1 µg/ml). Nuclear speckles were labeled with α-SRRM2 antibodies. *SRSF5-PCE* isoforms were labeled by RNA FISH. White arrowheads point to sites of SRSF5-PCE transcript accumulations. Scale bars = 5 µm.

We found that *SRSF5-PCE* transcripts accumulated in a bright spot that colocalized with SRSF5 and the nuclear speckle marker SRRM2 (Fig. 6 I). Interestingly, after 6 h of SRSF5 depletion, the bright *SRSF5-PCE* transcript spots were gone, and colocalization with nuclear speckles was no longer observed (Fig. 6 I). These data suggest that acute SRSF5 degradation may release *SRSF5-PCE* transcripts from nuclear speckles and destabilize them, while normal SRSF5 isoforms are not affected and thus their relative levels increase. Based on the rapid decline of many other non-coding transcripts at 2 h after SRSF5 depletion, the preferential binding of SRSF5 to downregulated transcripts without gross splicing alterations, we propose that SRSF5 binding protects them from decay.

Our study demonstrates the usefulness of hGRAD to study the role of nuclear condensates in gene expression and discover novel functions of nuclear RBPs by distinguishing direct from indirect or compensatory effects.

## Discussion

Studying the functions of individual RBPs is a challenging task. This is due to similar domain structure, redundant functions, cooperation and competition for binding sites, and large networks of auto- and crossregulation (Leclair et al., 2020; Meinke et al., 2020; Sliskovic et al., 2022). Moreover, their high mRNA translation rates and protein stability require long depletion times when using RNAi, which causes indirect effects (Chen et al., 2018). To discriminate between the functions of individual SR proteins, we and others have used endogenous GFP tagging by CRISPR/Cas9. Here, we introduce a simple system that allows the inducible, rapid, and efficient depletion of such GFP-tagged RBPs through genomic integration of a single plasmid. We show that our system, termed hGRAD, can deplete various GFP-tagged proteins expressed in different cell types and that localize to different subcellular compartments, including nuclear condensates such as nuclear speckles and paraspeckles. hGRAD is more efficient, less toxic, and requires less extensive genome editing than comparable approaches like the AID and Trim-Away systems.

hGRAD is derived from the deGradFP system, which was designed for degradation of GFP-tagged proteins in *Drosophila* (Caussinus et al., 2011). Several reports indicated that it is not suitable for the depletion of nuclear proteins in mammalian cells (Daniel et al., 2018; Li et al., 2019). To improve this, we exchanged the F-box domain from *Drosophila* Slmb for that of its

human ortholog FBXW11. A similar approach named zGRAD was shown to strongly increase degradation efficiency in zebrafish (Yamaguchi et al., 2019). hGRAD achieved a similar degradation efficiency compared with deGradFP and a significantly faster depletion than zGRAD.

Compared with the AUX-inducible systems, osTIR1 and aaAFB2, and Trim-Away, hGRAD did not impair cell proliferation upon induction. This is likely because the other approaches require the constitutive expression of large (70–90 kDa) heterologous E3-Ub ligases to be efficient (Clift et al., 2017; Daniel et al., 2018; Li et al., 2019), which contain additional enzymatic domains, making potential off-target effects more likely. The osTIR1 and aaAFB2 systems showed a faster onset of depletion, with half-lives down to 25 min (Daniel et al., 2018), because all required protein components are already expressed prior to induction with AUX. Moreover, a fast recovery of the target protein can be achieved after AUX removal (Daniel et al., 2018; Li et al., 2019). hGRAD had a longer onset and did not allow target protein recovery similar to the deGradFP and zGRAD systems (Caussinus et al., 2011; Yamaguchi et al., 2019). This might be due to the high stability of the nanobody fusions and the single layer of induction via DOX addition. However, the osTIR1 system was intrinsically leaky and led to significant degradation of SRSF3 prior to AUX addition, supporting other reports (Li et al., 2019). Moreover, the continuously expressed AID-nanobody possibly binds to GFP-tagged proteins and alters or inhibits their functions, which might contribute to the observed cytotoxic side effects.

While the deGradFP and zGRAD systems showed higher degradation efficacy in the cytoplasm (Caussinus et al., 2011; Yamaguchi et al., 2019), hGRAD degraded nuclear RBPs more efficiently. In the nucleus, hGRAD also outperformed the osTIR1, aaAFB2, and TRIM21 systems. It remains possible that these systems perform better for non-RBPs or cytoplasmic proteins, which was not tested here. The superior efficacy of hGRAD for nuclear RBPs may be explained, at least in parts, by the comparably small size of the F-box-nanobody fusions with an approximate size of 35–40 kDa, which allows their passive diffusion into the nucleus. Consistently, they were detected equally in both compartments. In addition to diffusion, it was also shown that the distribution of F-box proteins to different subcellular compartments can be promoted by binding to other factors (Kipreos and Pagano, 2000). In the cytoplasm, we observed a reduced depletion efficiency of hGRAD. This might result from the lower expression levels of the cytoplasmic

marker proteins used here (EIF4E and SRPK1), which could make depletion less visible. Additionally, it might reflect compartmental differences of the UPS. Nucleus and cytoplasm harbor exclusive sets of UPS proteins (Adori et al., 2006; Lafarga et al., 2002), and the distribution of the UPS in the nucleus or cytoplasm can vary depending on the cell type, development, and growth conditions (von Mikecz, 2006).

Using hGRAD, we were able to almost completely deplete the nuclear speckle protein SRSF5 after 3 h, which otherwise requires long depletion times by RNAi. This allowed us to study its immediate effects with minimized compensation and cross-regulation. The combination of hGRAD with Nascent-seq uncovered highly dynamic transcript changes after SRSF5 depletion. Interestingly, these changes were transient in nature and already appeared much attenuated at 16 h, which illustrates the robustness of gene expression through compensatory mechanisms likely by other SR proteins. Indeed, we discovered that SRSF5 depletion alters the inclusion levels of its own PCEs as well as that of other SR proteins, indicating that auto- and crossregulation operate when SRSF5 levels are too low. Moreover, our data suggest a novel role for SRSF5 in transcript stabilization, whereby SRSF5 binding targets and stores *SRSF5-PCE* transcripts and likely other non-coding transcripts in nuclear speckles. Acute depletion of SRSF5 releases bound transcripts from these condensates whereby they are exposed to nuclear decay. At later timepoints, other SR proteins might compensate for nuclear speckle targeting so that transcript levels recover.

## Conclusion

Rapid depletion is the tool of choice to study the functions of individual RBPs. hGRAD now allows the rapid degradation of notoriously difficult-to-target nuclear RBPs that localize to condensates. It allows disentangling protein functions from those of transcripts that are produced from the same genomic locus, to study auto- and crossregulation and to investigate the role of condensates in the regulation of gene expression. Our one-fits-all system can be inserted by a one-step procedure in any cell expressing GFP-tagged proteins, often commercially available and functionally validated. Moreover, the hGRAD vector is flexible and the F-box domain could be replaced with other targeting moieties or enzymatic activities, e.g., APEX2 for proximity labeling or kinases to modify GFP-tagged proteins in a rapid and inducible manner, making it a very useful tool for future studies. However, more work is required to further assess potential off-target effects of hGRAD.

## Materials and methods
### Generation of plasmid constructs
The pTRE-BI-hGRAD, -TRIM21, -aaAFB2, and -osTIR1 plasmids were generated by Gibson assembly (GBA). For this, a DOX-inducible bidirectional pTRE-BI vector was designed and synthesized by VectorBuilder (Vector ID: VB190904-1039fwc). For the pTRE-BI-hGRAD plasmid, a sequence encoding mCherry was added on one side of the Tet-inducible bidirectional promoter by conventional restriction cloning (*Bam*HI & *Spe*I; New England Biolabs). The coding region of a human F-box domain (215 amino

acids [aa], UniProt ID: Q9UKB1) fused to the sequence of an anti-GFP nanobody (VHH-GFP4; Saerens et al., 2005) was inserted on the other side by GBA. For the pTRE-BI-TRIM21 plasmid, the pTRE-BI vector was linearized with *Bam*HI and *Nhe*I (New England Biolabs), and inserts were integrated by GBA. The sequences encoding TRIM21 (476 aa, UniProt ID: P19474) fused to mCherry and a weak NLS signal (AAAKRVKLD; Li et al., 2019) was inserted at one side of the Tet-inducible bidirectional promoter and the human IgG1-Fc2 domain was inserted at the other side. For pTRE-BI-osTIR1, the *osTIR1* gene (574 aa, UniProt ID: Q7XVM8) fused to mCherry and a weak NLS signal was inserted at one side of the bidirectional promoter, and the mAID sequence (Daniel et al., 2018) and the GFP nanobody (VHH-GFP4) were added to the other side. For pTRE-BI-aaAFB2, the *aaAFB2* gene (575 aa, UniProt ID: Q9LW29) fused to mCherry and a weak NLS signal was inserted at one end of the bidirectional promoter, and the miniIAA7 AID sequence (Li et al., 2019) and the GFP nanobody (VHH-GFP4) to the other side.

Plasmids used as circular homology-directed repair (HDR) templates were generated by GBA. 1-kb homology arms flanking the guide RNA (gRNA) cut sites were amplified by PCR from host genomic DNA with primers containing overlapping sequences. The GFP-Neo resistance cassette containing a GFP tag, the Neomycin (Neo) resistance gene, and an internal ribosomal entry site was amplified from the SRSF3 BAC DNA (Müller-McNicoll et al., 2016). PCRs were performed with either S7 Fusion Polymerase (Mobidiag), Q5 High-Fidelity, or Taq DNA Polymerases (both New England Biolabs). PCR inserts and linearized pGEM-T Easy vector (Promega) were combined by GBA performed with Taq DNA Ligase and T5 Exonuclease (both New England Biolabs). One Shot TOP10 chemically competent *Escherichia coli* cells (Invitrogen) were used for transformations. Plasmids were extracted from bacteria with the ZR Plasmid Miniprep Classic (Zymo Research). Successful cloning was confirmed by Sanger sequencing (ACGT). All plasmids are listed in Table S4 and all primers in Table S7. The final plasmids have been deposited at Addgene: pTRE-BI-hGRAD (#207837; Addgene), pTRE-BI-TRIM21 (#207838; Addgene), pTRE-BI-aaAFB2 (#207839; Addgene), and pTRE-BI-osTIR1 (#207840; Addgene).

### Generation of stable cell lines and drug treatments
For integration of the pTRE-BI vectors, WT HeLa and P19 cells were transfected with 0.8 µg purified and linearized plasmid DNA per well in 6-well plates using the jetPRIME Transfection reagent (Polyplus-transfection). Cells with stably integrated plasmids were selected with 2 µg/ml puromycin (Gibco; Thermo Fisher Scientific). Single-cell clones were generated by limited dilution, grown from single cells in 96-well plates, and screened for high inducibility by confocal microscopy upon DOX induction (1 µg/ml).

BAC harboring sequences encoding for C-terminally GFP-tagged *SRSF3*, *SRSF5*, *NONO*, *EIF4E*, and *SRPK1* genes (Table S5) were isolated from *E. coli* DH10 cells using the NucleoBond Xtra Midi EF kit. For integration of the BACs, WT HeLa and P19 cells were transfected with 1 µg purified BAC DNA per well in 6-well plates using jetPRIME. Cells with stably integrated BACs were selected with 400 µg/ml Geneticin (G418; Gibco), sorted for

single-cell clones with near-endogenous expression levels, and expanded. All stable cell lines are listed in Table S6.

HeLa and P19 cells were cultivated under humidified conditions at 5% $CO_2$ and 37°C in DMEM GlutaMAX Medium, supplemented with 10% (vol/vol) heat-inactivated fetal bovine serum and 100 µg/ml penicillin-streptomycin (all Gibco, Thermo Fisher Scientific).

Live cells were counted with an EVE Automated Cell Counter device (NanoEnTek Inc). For this, equal volumes of cell suspension and 0.4% (vol/vol) trypan blue stain solution (Gibco, Thermo Fisher Scientific) were mixed and applied to EVE Cell counting slides. To inhibit the proteasome, HeLa cells were grown on coverslips placed in 24-well plates and treated with 10 µM of MG132 (M7449-200UL; Sigma-Aldrich) diluted in fresh DMEM for 4 h followed by fixation with 4% PFA (Thermo Fisher Scientific). To induce expression of hGRAD and TRIM21 systems, cells were treated with 1 µg/ml DOX (Sigma-Aldrich, D9891). osTIR1- and aaAFB2-expressing cells were additionally treated with 0.5 mM AUX (I2886-25G; Sigma-Aldrich).

### Genome editing with CRISPR/Cas9
For endogenous C-terminal tagging of *SRSF3*, *SRSF5*, and *SRRM2*, gRNAs were designed using CRISPOR (http://crispor.tefor.net) and purchased from Integrated DNA Technologies (IDT). HeLa and P19 cells were first transfected with circular HDR donor plasmids comprising the GFP-Neo resistance cassette flanked by gene-specific 1-kb homology arms using jetPRIME (Polyplus). After 6–12 h, WT cells were transfected with preassembled gRNAs (Alt-R CRISPR-Cas9 crRNA [target-specific, see Table S2], Alt-R CRISPR-Cas9 tracrRNA Atto550 labeled, IDT), and recombinant Cas9 protein (Alt-R *S. p.* HiFi Cas9 Nuclease V3; IDT) using Lipofectamine CRISPRMAX (Invitrogen) and cultured for 48 h in the presence of an HDR enhancer (final concentration 20 µM; IDT). Cells were then selected with Geneticin (400 µg/ml; Gibco). CRISPR clones were generated by limited dilution and grown from single cells in 96-well plates. For genomic screening, cells were washed twice with PBS in the 96-well plates and lysed in directPCR buffer (20 mM Tris-HCl, pH 8, 200 mM NaCl, 1 mM EDTA, 0.5% Tween-20, 0.5% NP-40) with freshly added 200 µg/ml proteinase K (Sigma-Aldrich), and incubated for 1 h at 55°C followed by proteinase K inactivation at 95°C for 15 min. Screening PCRs were performed from crude lysates using primers flanking the edited region. Sequences of the gRNAs are shown in Table S2 and primers in Table S7.

### RNA isolation, reverse transcription, and qPCR
Cells were harvested and resuspended in TRIzol (Thermo Fisher Scientific). RNA was extracted according to the manufacturer's instructions, treated with TURBO DNase (Thermo Fisher Scientific) for 30 min at 37°C to remove genomic DNA, and subsequently purified. 2 µg of RNA were reverse transcribed into cDNA using SuperScript and 10 mM dNTP Mix (both Thermo Fisher Scientific) and oligodT (Sigma-Aldrich). qPCR primers were selected using Primer-BLAST (https://www.ncbi.nlm.nih.gov/tools/primer-blast/). qPCRs were performed using cDNA (1:8 dilution) and the ORA SEE qPCR Green ROX L kit (highQu) on a PikoReal 96 machine (Thermo Fisher Scientific). GraphPad

Prism was used for all graphics. Primers used are listed in Table S7.

### Western blot and antibodies
Cells were lysed in 300 µl NET-2 buffer (150 mM NaCl, 0.05% [vol/vol] NP-40, 50 mM Tris-HCl, pH 7.5), supplemented with 1× cOmplete Protease Inhibitor Cocktail (Sigma-Aldrich) and 10 mM β-phosphoglycerate (Fluka BioChemica) or with radio-immunoprecipitation assay (RIPA) buffer (150 mM NaCl, 0.05% [vol/vol] NP-40, 50 mM Tris-HCl, pH 7.5, 0.1% [wt/vol] SDS, 0.5% [wt/vol] sodium deoxycholate, freshly added 1× cOmplete Protease Inhibitor Cocktail and 10 mM β-phosphoglycerate). NET-2 lysates were sonicated on ice for 30 s (three pulses of 10 s; 20-s intervals) at 20% amplitude (Branson W-450 D) and cleared by centrifugation. Protein concentrations were measured using Quick Start Bradford 1× Dye Reagent (Bio-Rad) on a NanoDrop 2000 (Thermo Fisher Scientific) or DC Protein-Assay (Bio-Rad) for RIPA samples. 20–40 µg protein were separated by SDS-PAGE on 4–15% Mini-PROTEAN TGX Stain-Free Gels (Bio-Rad) and transferred onto polyvinyliden fluoride (PVDF) membranes using Trans-Blot Turbo RTA Mini LF PVDF Transfer Kit (Bio-Rad). Transfer and equal loading were evaluated by activation of stain-free gels by UV light. Membranes were probed with the antibodies listed in Table S3. Proteins were imaged using secondary antibodies coupled to a horseradish peroxidase and Amersham ECL Prime Western Blotting Detection Reagent (Cytiva) with the ChemiDoc MP Imaging System. Image quantification was performed using the ImageLab software (Bio-Rad).

### RNA-FISH-immunofluorescence (IF), GFP fluorescence microscopy, and live-cell imaging
For RNA-FISH-IF experiments, 12-mm coverslips were placed inside the wells of a 24-well plate. Cells were fixed with 4% PFA in PBS for 10 min and permeabilized with 70% ethanol overnight. FISH was performed using Stellaris probes and buffers (LG Biosearch Technologies) following the manufacturer's protocol. Coverslips were washed with Stellaris Wash Buffer A, placed in a humidified chamber, and hybridized for 16 h at 37°C in the dark with *SRSF5*-Cy5 FISH probes and rabbit α-SRRM2 antibody (PA5-59559) as nuclear speckle marker, both diluted 1:100 in Stellaris hybridization buffer. After hybridization, the coverslips were incubated with Stellaris Wash Buffer A containing the secondary antibody (donkey α-rabbit coupled to Alexa Fluor 405, A48258) in a 1:500 dilution for 30 min at 37°C. DNA was stained with Hoechst 33342 (Sigma-Aldrich) at a final concentration of 5 µg/ml in Wash Buffer A for 30 min at 37°C. Coverslips were washed, dried, and mounted onto glass slides using ProLong Diamond Antifade Mountant (P36961; Thermo Fisher Scientific).

For GFP fluorescence microscopy, cells were grown on 10-mm glass coverslips. After removing the medium, cells were fixed with 4% PFA (Thermo Fisher Scientific) in PBS for 10 min at room temperature. DNA was stained with Hoechst 34580 (1:4,000; Thermo Fisher Scientific). After a final wash, the coverslips were dried and mounted on ProLong Diamond Antifade Mountant (P36961; Thermo Fisher Scientific).

For live-cell imaging, cells were grown on a microscopy grade 24-well glass-bottom culture plate (Sensoplate; Greiner Bio-ONE). Prior to imaging, the medium was exchanged with low autofluorescent FluoroBrite DMEM Media (Gibco) supplemented with 10% (vol/vol) FBS, 2 mM GlutaMAX I Supplement, and 1 mM sodium pyruvate (all Gibco, Thermo Fisher Scientific). 100 nM SiR-DNA stain (Spirochrome) was added for the visualization of nuclei.

## Image acquisition and quantification

Images were acquired with a confocal laser-scanning microscope (LSM780; Zeiss) with a Plan-Apochromat 63× 1.4 NA oil differential interference contrast objective M27 using the Zen 2012 (black edition; 8.0.5.273; ZEISS). Fluorescence signal was detected with an Argon laser (GFP-488 nm, Qasar 570–561 nm, and Qasar 670–647 nm). Images from the same experiment were acquired on the same day with the same settings for all conditions. Pictures were cropped with the image crop function and scale bars were added.

For live-cell imaging, cell culture plates were mounted on the Zeiss LSM 780 microscope, equipped with a humidified incubation chamber (preheated for 2 h at 37°C, 5% $CO_2$). Selected regions were automatically imaged every 15–20 min with preset settings and software autofocus using Zen 2012 (black edition; 8.0.5.273; ZEISS) software. GFP intensity and SiR-DNA were visualized in parallel. Eight Z-stacks per region with a slice thickness of 2 μm were imaged. To minimize phototoxicity, minimal possible laser intensity was chosen (488 nm <3%, 670; 633 nm <5%). GFP intensity changes in individual cells were quantified over time using Fiji software (Schindelin et al., 2012) and the TrackMate plugin (Tinevez et al., 2017). First, a Z-maximum intensity projection was made to generate a single-plane image. The background was subtracted using a rolling ball background subtraction algorithm with a disk size of 75 μm. TrackMate was first applied to the SiR-DNA channel to identify and connect cell tracks over time. The mean GFP intensity was evaluated in corresponding cells and normalized to the first timepoint.

## 4sU treatment and Nascent-seq

HeLa SRSF5-endo-GFP cells were induced with DOX. After 2 h (T2), 8 h (T8), and 16 h (T16), 400 μM 4sU was added for 1 h in the presence of DOX to label newly transcribed RNAs. Uninduced cells were used as controls (T0). The labeling reaction was stopped by adding TRIzol reagent to the culture dishes. RNA extraction and pulldown of newly transcribed RNA were performed according to Gressel et al. (2019) without fractionation of the RNA by sonication. Nascent RNA was purified using RNA Clean and Concentrate Kit (ZymoResearch). cDNA libraries for RNA sequencing were prepared with universal Plus Total RNA-Seq Library Preparation Kit (Tecan) according to manufacturer's instructions. Ribosomal RNA fragments were removed and the library was sequenced on an Illumina NovaSeq 6000 instrument (two replicates per timepoint, 100 million reads, 150 bp, paired-end). Reads were mapped to the human genome (hg38) using STAR (version 2.7.10; Dobin et al., 2013) with the following parameters: --outFilterMultimapNmax 1 --outFilterMismatchNmax 999 --outFilterMismatchNoverReadLmax 0.04 --outSAMtype BAM SortedByCoordinate.

## Quantification of differential expression, alternative splicing, and intron retention

To globally quantify the alternative splicing, MAJIQ (version 2.3; Jha et al., 2017) was run. Initially, a splice graph was built using the BAM files from all conditions and GENCODE gene annotation (version 38, genome version GRCh38) using MAJIQ build. Next, the difference in junction usage for timepoints T2, T8, and T16 compared with T0 was calculated (as ΔPSI) using MAJIQ deltapsi. Subsequently, alternative splicing events, e.g., cassette exons and IR, were identified and quantified using MAJIQ Modulizer (Vaquero-Garcia et al., 2023) with the following parameters: --changing-between-group-dpsi 0.05 --non-changing-between-group-dpsi 0.02 --changing-between-group-dpsi-secondary 0.025 --show-all. To filter for significantly regulated events, the MAJIQ Modulizer output was further processed in R: Splicing events with two junctions were defined as significantly regulated if, for both junctions, probability changing (Ps) ≥0.9 and |ΔPSI| ≥0.055. Events with four junctions were defined as significantly regulated, if Ps ≥0.9 for at least one junction pair (inclusion + skipping junction) of the same local splice variants, |ΔPSI| ≥0.055 for all junctions, within junction pairs the lower |ΔPSI| is at least 50% of the higher |ΔPSI|, and the sign within both junction pairs is inverse. To compare significantly regulated IR events between timepoints, IR events were filtered such that each event was fully quantified for each timepoint and was significantly regulated at least one timepoint. In total, this resulted in 121 significantly regulated IR events.

To further quantify IR events, IRFinder (version 1.3.1; Middleton et al., 2017) was used with default parameters. Differential gene expression was quantified using DESeq2 with default parameters (Love et al., 2014). Sashimi plots for PCE of selected SR proteins were prepared using the R/Bioconductor package Gviz (version 1.37.2) by taking the merged BAM files of each timepoint as input. Junctions were filtered for those associated with the inclusion levels of the PCE.

## iCLIP2 libraries

iCLIP experiments were performed using the iCLIP2 protocol (Buchbender et al., 2020) with minor modifications. For each replicate, cells were grown near confluence on two 150-mm culture dishes, washed with ice-cold PBS, irradiated with 150 mJ/cm² UV light at 254 nm (CL-1000; UVP), harvested by scraping and centrifugation and stored at –80°C until lysis. Following lysis and partial digestion with RNase I (AM2294; Thermo Fisher Scientific), immunoprecipitation of SRSF5-GFP was performed using a goat anti-GFP antibody (MPI-CBG) coupled to Dynabeads Protein G (10002D; Thermo Fisher Scientific). Copurified, crosslinked RNA fragments were dephosphorylated at their 3′ ends using T4 Polynucleotide Kinase (M0201S; New England Biolabs) and ligated to a pre-adenylated 3′ adapter (L3-App). To visualize protein–RNA complexes, RNA fragments crosslinked to SRSF5-GFP were labeled at their 5′ ends using T4 Polynucleotide Kinase and γ-³²P-ATP (Hartmann Analytic). Samples were run on a Nu-PAGE 4–12% Bis-Tris Protein Gel

(NP0335BOX; Thermo Fisher Scientific), transferred to a 0.45 µm nitrocellulose membrane (10600002; GE Healthcare Life Science), and visualized using a Phosphorimager. Regions of interest were cut from the nitrocellulose membrane (70–130 kDa) and RNA was released from the membrane using proteinase K (03115828001; Roche). RNA was purified using neutral phenol/chloroform/isoamylalcohol (AM9722; Ambion) followed by chloroform (39554.02; Serva) extraction and reverse transcribed using SuperScript III (18080-044; Life Technologies). cDNA was cleaned up using MyONE Silane beads (37002D; Life Technologies) followed by ligation of a second adapter containing a bipartite (5 + 4-nt) unique molecular identifier (UMI) as well as a 6-nt experimental barcode (Buchbender et al., 2020). iCLIP2 libraries were preamplified with six PCR cycles using short primers (P5Solexa_short and P3Solexa_short) and then size-selected using the ProNex Size-Selective Purification System (NG2001; Promega) in a 1:2.95 (vol/vol) sample:bead ratio to eliminate products originating from short cDNAs or primer dimers. The size-selected library was amplified for six cycles using P5Solexa and P3Solexa primers and primers were removed using the ProNex Size-Selective Purification System (NG2001; Promega) in a 1:2.4 (vol/vol) sample:bead ratio. Purified iCLIP2 libraries were sequenced on a NextSeq 500 System (Illumina) using a NextSeq 500/550 High Output Kit v2 as 92-nt single-end reads, yielding between 5.4 and 74.1 million reads.

### iCLIP2 analysis

Basic quality controls were done using FastQC (version 0.11.8; https://www.bioinformatics.babraham.ac.uk/projects/fastqc/). The FASTX-Toolkit (version 0.0.14) and seqtk (version 1.3; https://github.com/lh3/seqtk/) were used to filter reads based on sequencing qualities (Phred score) in the barcode and UMI regions. Reads were demultiplexed according to the sample barcode on positions 6–11 of the reads using Flexbar (version 3.4.0, using non-default parameter --barcode-keep; Roehr et al., 2017). Flexbar was also used to trim UMI and barcode regions as well as adapter sequences from read ends requiring a minimal overlap of 1 nt of read and adapter. UMIs were added to the read names, and reads <15 nt were removed from further analysis. The downstream analysis was done as described in Chapters 3.4 and 4 of Busch et al. (2020) with an additional step to remove reads directly mapped to the chromosome ends using Samtools (version 1.9; Danecek et al., 2021) and bedtools (version 2.27.1; Quinlan and Hall, 2010). Those reads do not have an upstream position and, thus, no crosslink position can be extracted. Genome assembly and annotation of GENCODE (release 31; Frankish et al., 2019) were used during mapping with STAR (version 2.7.3a; Dobin et al., 2013).

Processed reads from five replicates were merged prior to peak calling with PureCLIP (version 1.3.1; Krakau et al., 2017) using a minimum transition probability of 1%. Significant crosslink sites (1 nt) were filtered by their PureCLIP score, removing the lowest 2% of crosslink sites. The remaining sites were merged into 7-nt-wide binding sites using the R/Bioconductor package BindingSiteFinder (version 1.0.0), filtering for sites with at least three positions covered by crosslink events. Only reproducible binding sites were considered for further analyses, which had to be supported by four out of five replicates. Binding sites were overlapped with gene and transcript annotations obtained from GENCODE (release 29). Binding sites within protein-coding genes were assigned to the transcript regions, i.e., intron, coding sequence, 3′UTR, or 5′UTR.

### Online supplemental material

Fig. S1 shows that hGRAD performs superior to other inducible degradation systems. Fig. S2 shows that hGRAD works in different cell types and species. Fig. S3 shows that hGRAD efficiently degrades endogenously GFP-tagged nuclear RBPs. Fig. S4 shows that hGRAD efficiently degrades endogenously GFP-tagged nuclear RBPs in P19 cells allowing their rapid knockdown. Fig. S5 shows that combining hGRAD with Nascent-seq allows the identification of direct SRSF5 targets. Table S1 lists integration of DESeq2, IR-Finder, and iCLIP2. Table S2 provides the list of gRNAs. Table S3 provides the list of antibodies used in this study. Table S4 lists the plasmids used or generated in this study. Table S5 lists the BACs used in this study. Table S6 provides the list of cell lines used or generated in this study. Table S7 lists the primers used in this study.

### Data availability

The plasmids for hGRAD, TRIM-away, osTIR, and aaAFB2 are available from Addgene. All sequencing data is available in the Gene Expression Omnibus (GEO) under the SuperSeries accession number GSE229326. The collection includes the Nascent-seq data of SRSF5-GFP-endo (GSE229324) as well as the iCLIP data for SRSF5-GFP-endo in HeLa cells (GSE229325). All other data are available from the corresponding author upon reasonable request.

## Acknowledgments

We thank Maria Clara Hernandez Cañas for running IRFinder, Ina Poser (Max Planck Institute of Molecular Cell Biology and Genetics) for the GFP-tagged BACs, and Anke Busch and IMB Bioinformatics Core Facility for processing the iCLIP2 data.

Support by the IMB Genomics Core Facility and the use of its NextSeq 500 (funded by the Deutsche Forschungsgemeinschaft [DFG, German Research Foundation]—329045328) is gratefully acknowledged. We are grateful for funding from the Deutscher Akademischer Austauschdienst to I. Slišković and Deutsche Forschungsgemeinschaft (SFB902-B13 to B. Arnold, K. Zarnack, and M. Müller-McNicoll).

Author contributions: B. Arnold and M. Müller-McNicoll designed the experiments. B. Arnold, R.J. Riegger, F. McNicoll, and E.K. Okuda performed the experiments. B. Arnold, R.J. Riegger, I. Slišković, M. Keller, C. Bakisoglu, and K. Zarnack performed the analyses. Figures were prepared by B. Arnold, R.J. Riegger, and E.K. Okuda. The manuscript was written by B. Arnold, R.J. Riegger, K. Zarnack, and M. Müller-McNicoll.

Disclosures: The authors declare no competing interests exist.

Submitted: 12 April 2023

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

# Supplemental material

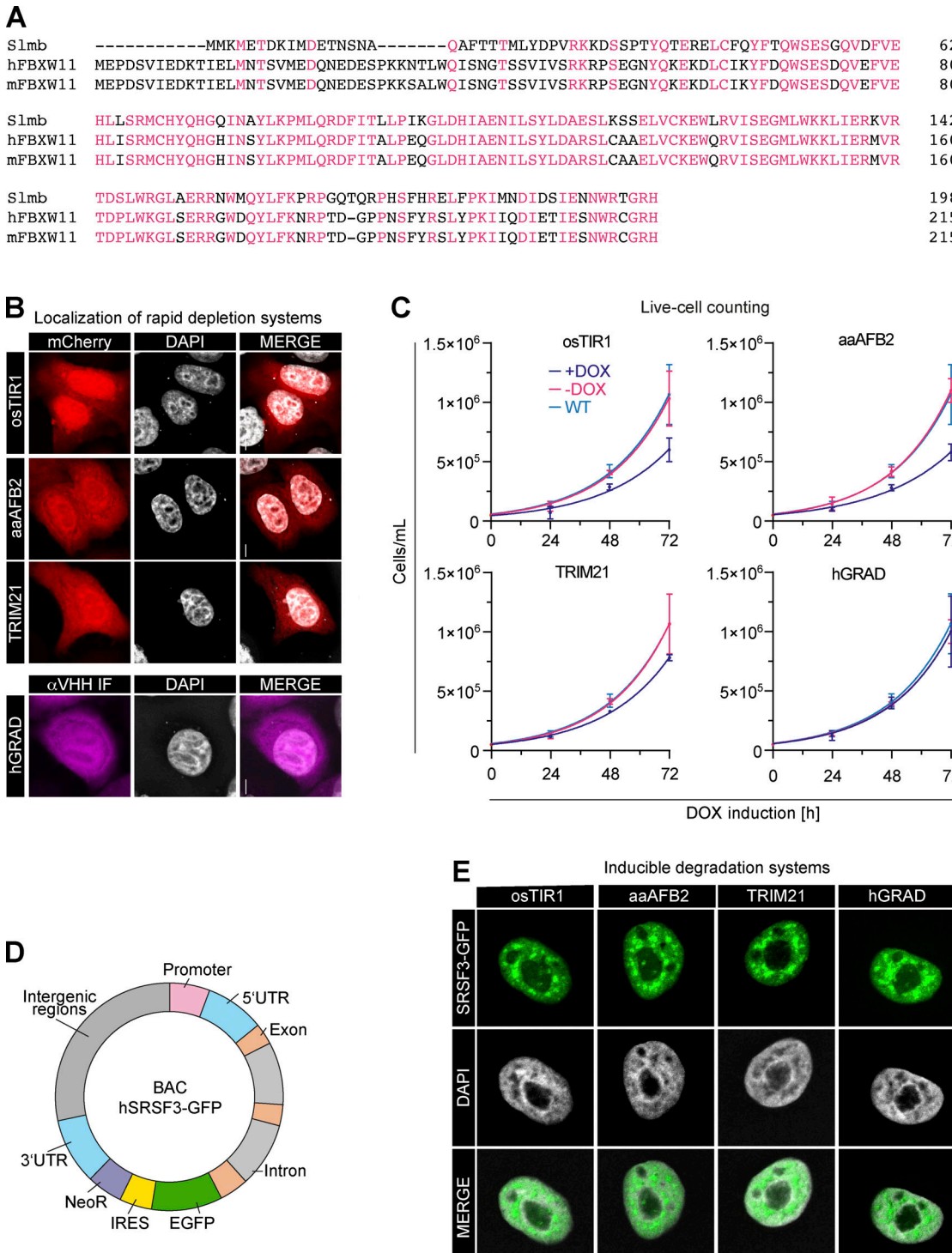

**Figure S1.** **hGRAD performs superior to other inducible degradation systems. (A)** Multiple sequence alignment of Slmb (*Drosophila*) and its orthologs hFBXW11 and mFbxw11. **(B)** Subcellular localization of the induced proteins. Microscopy images of different HeLa master cell lines induced with DOX (1 µg/ml) for 16 h. Upper panel: osTIR1-, aaAFB2-, and TRIM21-mCherry (red), DNA (gray). Lower panel: hGRAD F-box-nanobody fusion was detected with an anti-VHH antibody and an Alexa Fluor 647 dye-coupled secondary antibody (magenta), DNA (gray). Note that the osTIR1, aaAFB2, and TRIM21 systems include mCherry fusion proteins, whereas the hGRAD system uses the endogenous UPS. Hence, for hGRAD, mCherry only serves as an induction control (not shown), while the GFP nanobody was detected by IF against VHH. Scale bars = 5 µm. **(C)** Experimental scheme and growth curve comparison of HeLa live cells expressing the osTIR1, aaAFB2, TRIM21, or hGRAD systems until 72 h after DOX induction. The $t_d$ was evaluated by exponential growth equation fit ($Y=Y_0*exp(k*X)$). WT control (blue), uninduced control (pink), induced (purple). Mean and SD are shown from $n$ = 3 independent experiments. **(D)** Scheme of a BAC that was randomly integrated into the genome of HeLa cells. The BAC carries the complete human *SRSF3* gene fused to *EGFP* including the *SRSF3* promoter, 5′UTR, exons, introns, and 3′UTR to ensure near endogenous expression and isoform levels. IRES, internal ribosomal entry site; NeoR, Neomycin resistance gene. **(E)** SRSF3-GFP is expressed at similar levels in all degradation systems and localizes to nuclear speckles and the nucleoplasm in HeLa cells. Scale bars = 5 µm.

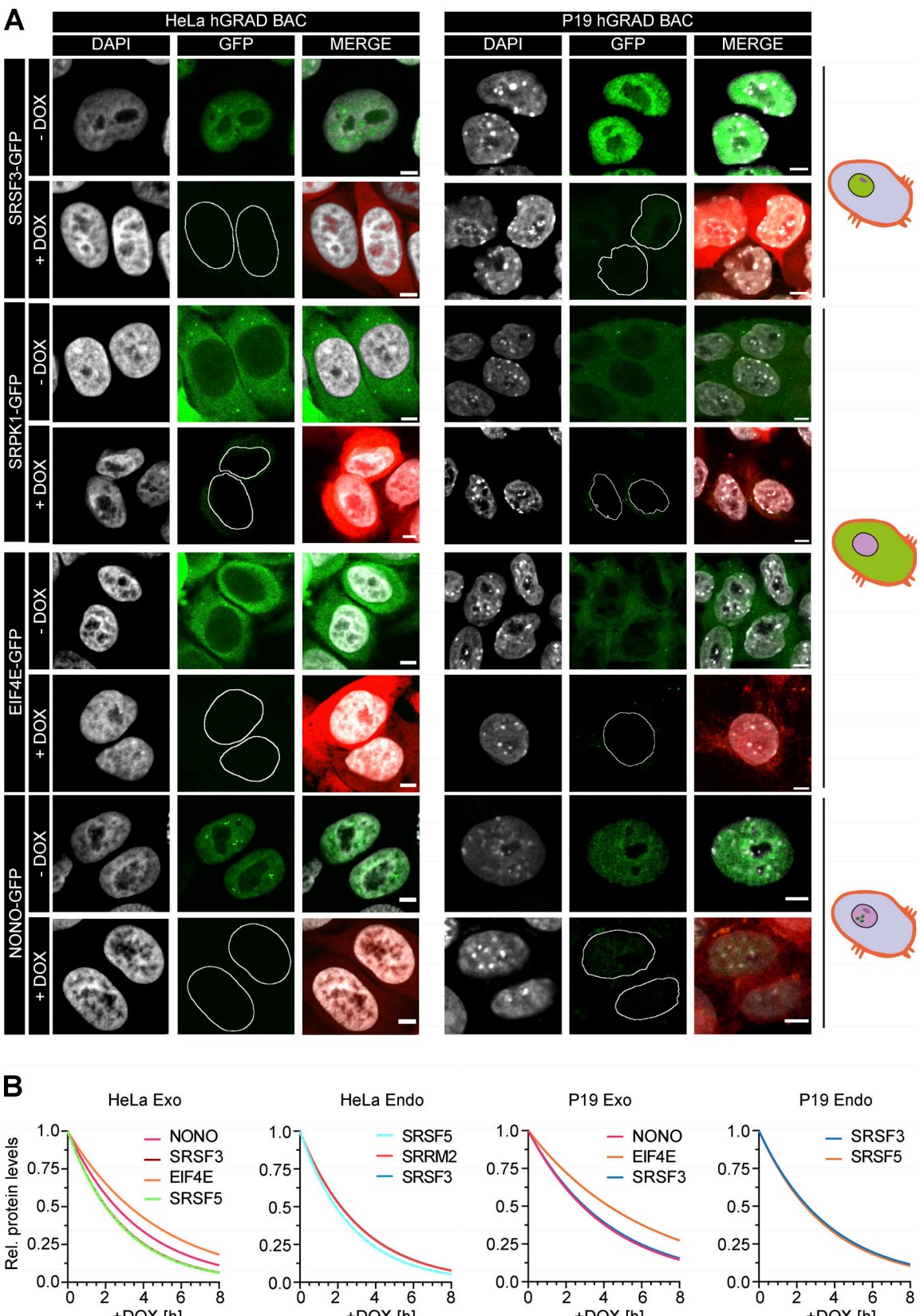

Figure S2. **hGRAD works in different cell types and species. (A)** Representative micrographs showing the subcellular localization and degradation of GFP-tagged SRSF3, SRPK1, EIF4E, and NONO in human HeLa and mouse P19 cells after 16 h induction by DOX (1 µg/ml). GFP-tagged proteins were expressed from integrated BACs. Red channel shows mCherry induction control. Scale bars = 5 µm. **(B)** Non-linear regression fit (Exponential One Phase Decay Model) to determine the protein half-lives of GFP-tagged proteins in HeLa and P19 cells based on western blots.

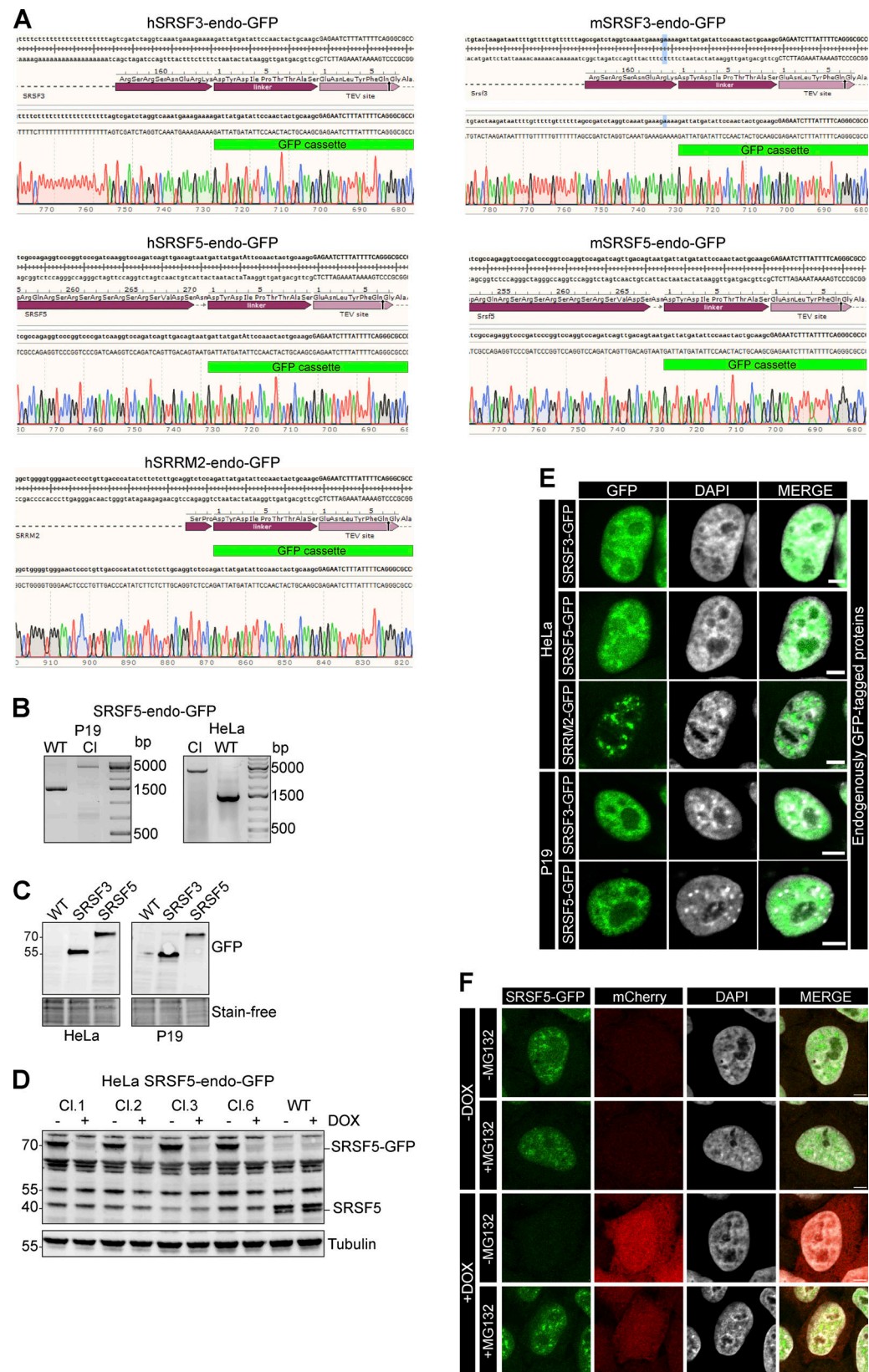

Figure S3. **hGRAD efficiently degrades endogenously GFP-tagged nuclear RBPs. (A–D)** Correct insertion of the GFP cassette in the genomic locus of human (hSRSF3, hSRSF5, and hSRRM2) and mouse (mSRSF3 and mSRSF5) RBP genes was validated (A) by Sanger sequencing, (B) by PCR amplification of the genomic locus, (C) by western blot using an anti-GFP antibody, and (D) by western blot using the anti-SRp40 (Sigma-Aldrich) antibody. Tubulin served as loading control. **(E)** Subcellular localization of GFP-tagged proteins was validated by confocal microscopy. **(F)** SRSF5 degradation is blocked when the proteasome is inhibited by MG132 (10 μM) for 4 h. Scale bars = 5 μm. Source data are available for this figure: SourceData FS3.

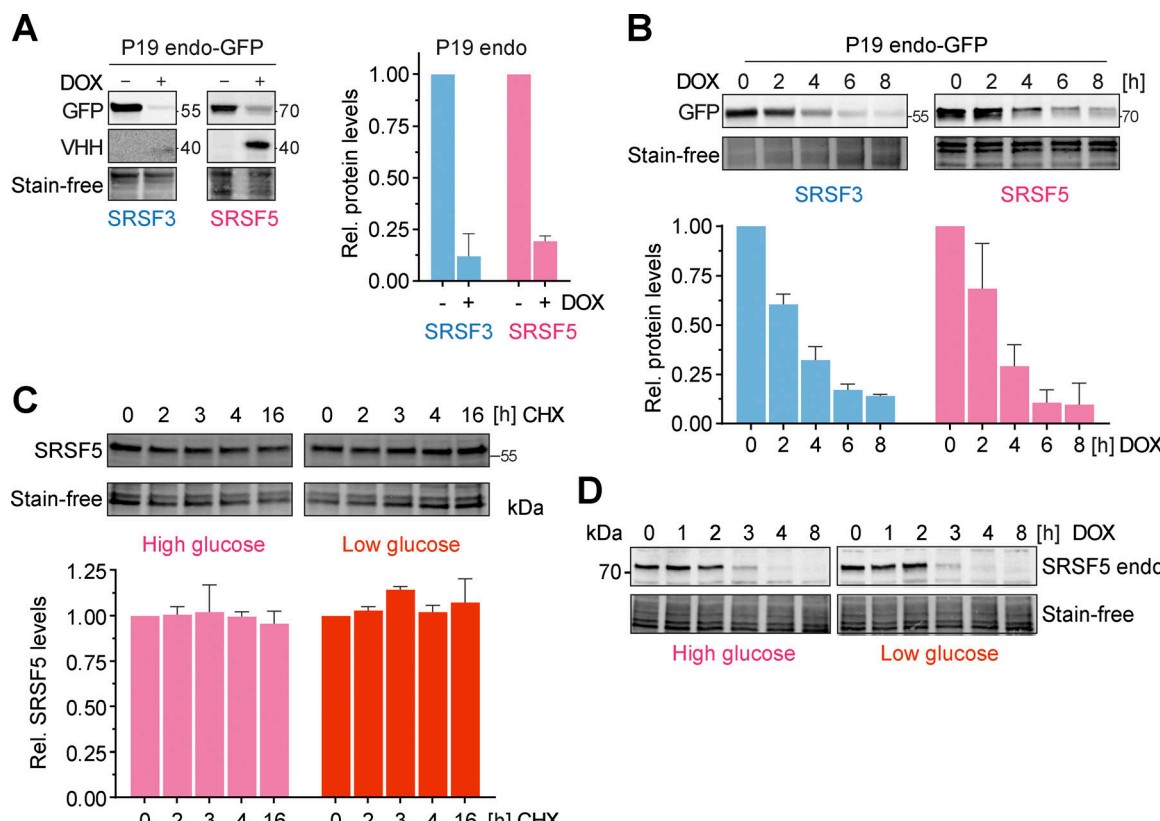

Figure S4. **hGRAD efficiently degrades endogenously GFP-tagged nuclear RBPs in P19 cells allowing their rapid knockdown. (A)** Degradation efficiency of SRSF3- and SRSF5-endo-GFP in P19 cells after 16 h DOX induction (1 µg/ml). **(B)** Degradation time courses show that hGRAD works efficiently for SRSF3- and SRSF5-endo-GFP in P19 cells. **(C)** Representative western blot for a time-course under high (4.5 g/L) and low (1.0 g/L) glucose conditions over 16 h upon treatment with the eukaryotic translation inhibitor cycloheximide (CHX; 10 µg/ml) to determine the stability of SRSF5-endo-GFP in HeLa cells. **(A–C)** Shown are one representative western blot and the quantification (mean and SD) of n = 3 independent experiments. Quantifications relative to the −DOX or 0 h timepoint. Stain-free membranes were used to control for equal loading. **(D)** Comparison of SRSF5-endo-GFP degradation by hGRAD in HeLa cells in high and low glucose conditions. Source data are available for this figure: SourceData FS4.

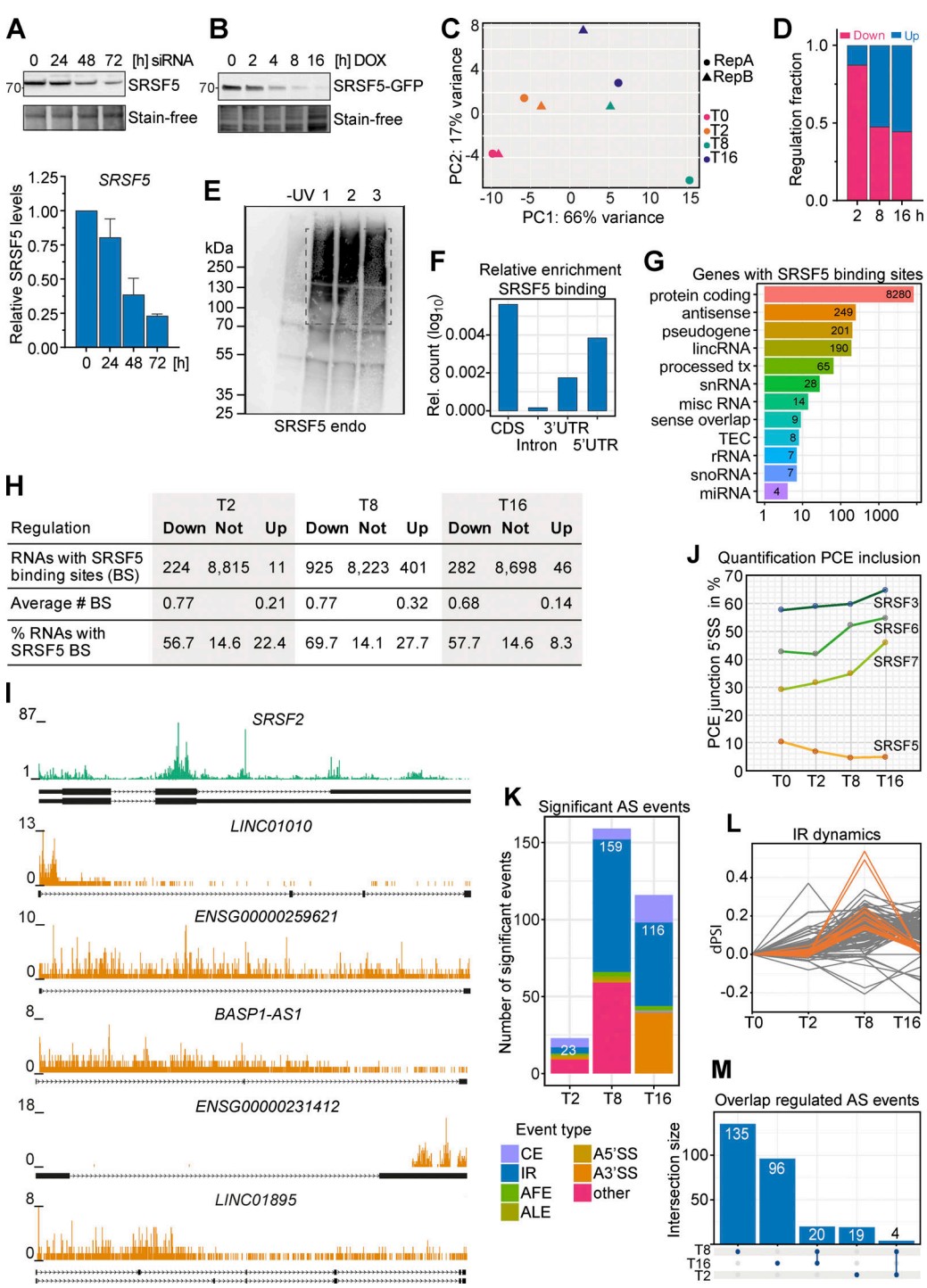

Figure S5. **Combining hGRAD with Nascent-seq allows the identification of direct SRSF5 targets. (A)** Top: Representative western blot for a SRSF5 knockdown using siRNAs in a 72-h time-course experiment. Bottom: Quantification (mean and SD) relative to the 0 h sample of $n = 2$ independent experiments. **(B)** Representative degradation time course of SRSF5-endo-GFP used for Nascent-seq. **(C)** Principal component analysis (PCA) shows the clustering of replicates in PC1 and 2. **(D)** Fraction of down- and upregulated transcripts at the timepoints T2, T8, and T16 after DOX induction. **(E)** A representative autoradiograph of SRSF5-endo-GFP cells used for iCLIP2 showing three out of five replicates. Non-crosslinked cells (−UV) were used as control. **(F)** Enrichment of SRSF5 binding sites in specific transcript regions, i.e., coding region (CDS), introns, and 3' and 5'UTR relative to feature length. **(G)** Genes with SRSF5 binding sites sorted by gene biotype. lincRNA, long intergenic noncoding RNA; tx, transcript; snRNA, small nuclear RNA; misc, miscellaneous; TEC, to be experimentally confirmed; rRNA, ribosomal RNA; snoRNA, small nucleolar RNA; miRNA, microRNA. **(H)** SRSF5 binding sites in regulated and non-regulated transcripts. **(I)** Genome browser views with SRSF5 crosslinks on *SRSF2* mRNA and selected lincRNAs that are downregulated after rapid SRSF5 depletion. **(J)** Percentage of junctions including the PCE of *SRSF3*, *SRSF5*, *SRSF6*, and *SRSF7* at T0–T16. **(K)** Significant alternative splicing (AS) events at T2, T8, and T16 quantified using MAJIQ. CE, cassette exon; AFE, ALE, alternative first or last exon; A5'SS, A3'SS, alternative 5' or 3' splice sites. Significance: dPSI >0.055; Probability >0.9. **(L)** Dynamics of IR events. dPSI, delta percent spliced in. **(M)** Upset plot showing the overlap of regulated AS events at T2, T8, and T16. Source data are available for this figure: SourceData FS5.

Provided online are seven tables. Table S1 lists the integration of DESeq2, IRFinder, and iCLIP2. Table S2 is a list of gRNAs. Table S3 lists the antibodies used in this study. Table S4 provides the list of plasmids used or generated in this study. Table S5 lists the BACs used in this study. Table S6 lists the cell lines used or generated in this study (selection marker: puromycin [Puro] or geneticin [Gen] as indicated). Table S7 lists the primers used in this study.

