## [Peer Review File · The Journal of Cell Biology]

hGRAD - a versatile 'one-fits-all' system to acutely deplete RNA binding proteins from condensates

Benjamin Arnold, Ricarda Riegger, Ellen Okuda, Irena Sliskovic, Mario Keller, Cem Bakisoglu, François McNicoll, Kathi Zarnack, and Michaela Müller-McNicoll

Corresponding Author(s): Michaela Müller-McNicoll, Goethe University Frankfurt

Review Timeline:

Submission Date:	2023-04-12
Editorial Decision:	2023-06-21
Revision Received:	2023-09-18
Editorial Decision:	2023-10-13
Revision Received:	2023-10-26

Monitoring Editor: Kenneth Yamada

Scientific Editor: Tim Spencer

Transaction Report:

DOI: <https://doi.org/10.1083/jcb.202304030>

June 21, 2023

Re: JCB manuscript #202304030

Prof. Michaela Müller-McNicoll
Goethe University Frankfurt
Institute for Molecular Biosciences
Max-von-Laue-Str. 13
Frankfurt am Main 60438
Germany

Dear Prof. Müller-McNicoll,

Thank you for submitting your manuscript entitled "hGRAD - a versatile 'one-fits-all' system for the acute depletion of RNA binding proteins in nuclear condensates". The manuscript was assessed by three expert reviewers, whose comments are appended to this letter. We invite you to submit a revised manuscript if you can address the reviewers' key concerns, as outlined here. In addition, we would like to point out that we (the Editors) felt that your manuscript is a better fit for the JCB Tools format (rather than as an Article), and the reviewers agree. This reclassification will not require you to reformat your paper in any way -- it simply alters the criteria that we and the reviewers use when assessing its suitability for JCB.

The description of the JCB Tools format is as follows:

"Tools describe new methods or datasets (e.g., screens, systems-wide analyses, or computational modeling) of immediate value and broad utility to the cell biology community. Papers presenting methods should describe a technological advance of broad/general interest that permits the interrogation of cell biological problems in ways previously impossible and include novel cell biological insight as a proof of principle. For datasets, authors must provide compelling proof of principle that analysis of the dataset yields novel cell biological insights."

In any case, you will see that while all three reviewers exhibit enthusiasm for the paper, they also raise a number of concerns that will need to be addressed before the paper would be suitable for publication. In particular, the reviewers feel that better controls are needed to support the conclusions indicating that the observed cellular phenotypes are due to SRSF5 depletion. Although Reviewer #1 also suggests that you remove the SRSF5-related data from the manuscript entirely, we disagree - this aspect of the paper nicely fulfills the 'proof of principle' requirement for providing a novel cell biological insight that is necessary for the Tools format. Therefore, we ask that you instead fully address the other reviewers' comments to reinforce this part of the paper. We also hope that you will be able to address each of the reviewers' other comments in full, with the exceptions outlined below.

Reviewer #3 requests that you examine whether modulation of SRSF5 alters alternative splicing. While this is an interesting avenue of examination, addressing this experimentally will not be required for the revision (though you may wish to consider adding discussion/speculation on this point). Reviewer #1 also suggests that you examine degradation of non-RBP proteins in the nuclear and cytoplasmic compartments. We agree with the reviewer that addressing this issue would further demonstrate the utility of the hGRAD system, and so we would encourage you to perform these experiments - however, we will not require it for resubmission.

GENERAL GUIDELINES:

Text limits: Character count for Tools articles is < 40,000, not including spaces. Count includes title page, abstract, introduction, results, discussion, and acknowledgments. Count does not include materials and methods, figure legends, references, tables, or supplemental legends.

Figures: Tools may have up to 10 main text figures. Figures must be prepared according to the policies outlined in our Instructions to Authors, under Data Presentation, <https://jcb.rupress.org/site/misc/ifora.xhtml>. All figures in accepted manuscripts will be screened prior to publication.

Supplemental information: There are strict limits on the allowable amount of supplemental data. Tools may have up to 5 supplemental figures. Up to 10 supplemental videos or flash animations are allowed. A summary of all supplemental material should appear at the end of the Materials and methods section.

Please note that JCB now requires authors to submit Source Data used to generate figures containing gels and Western blots with all revised manuscripts. This Source Data consists of fully uncropped and unprocessed images for each gel/blot displayed in the main and supplemental figures. Since your paper includes cropped gel and/or blot images, please be sure to provide one Source Data file for each figure that contains gels and/or blots along with your revised manuscript files. File names for Source Data figures should be alphanumeric without any spaces or special characters (i.e., SourceDataF#, where F# refers to the associated main figure number or SourceDataFS# for those associated with Supplementary figures). The lanes of the gels/blots should be labeled as they are in the associated figure, the place where cropping was applied should be marked (with a box), and molecular weight/size standards should be labeled wherever possible.

The typical timeframe for revisions is three to four months. While most universities and institutes have reopened labs and allowed researchers to begin working at nearly pre-pandemic levels, we at JCB realize that the lingering effects of the COVID-19 pandemic may still be impacting some aspects of your work, including the acquisition of equipment and reagents. Therefore, if you anticipate any difficulties in meeting this aforementioned revision time limit, please contact us and we can work with you to find an appropriate time frame for resubmission. Please note that papers are generally considered through only one revision cycle, so any revised manuscript will likely be either accepted or rejected.

Thank you for this interesting contribution to Journal of Cell Biology. You can contact us at the journal office with any questions, cellbio@rockefeller.edu.

Sincerely,

Kenneth Yamada, MD, PhD
Senior Editor
Journal of Cell Biology

Tim Spencer, PhD
Executive Editor
Journal of Cell Biology

Reviewer #1 (Comments to the Authors (Required)):

In this study, Arnold and colleagues developed a novel system for acute depletion of RNA-binding proteins (RBPs), which they termed, hGRAD. This novel system was adapted from the deGradFP system, which was developed to degrade GFP-tagged proteins in *Drosophila*. This degradation system relies on human FBP, FBXW11, a close orthologue to *Drosophila* Slmb, which is fused to an anti-GFP nanobody (VHH-GFP4) and induces degradation of GFP-tagged proteins via recruitment of the endogenous SCF complex.

The authors compare the hGRAD system with other degrading systems, including TRIM21, TIR1 and AFB2, which were cloned in the same vector and was initially tested for the degradation of GFP-SRSF3. The authors extended their analysis to the degradation of another SR protein family member, SRSF5 and to two additional RBPs, SRRM2 and NONO. They concluded that the hGRAD degrades nuclear RBPs more efficiently than the other tested systems.

Next, Arnold and co-authors show that hGRAD does indeed degrade RBPs that localise to nuclear condensates and this system is very efficient across different cell lines and species (HeLa, mouse P19). The authors went on to combine hGRAD-mediated degradation of SRSF5 with Nascent-seq, which due to the rapid degradation of SRSF5 allowed them to identify direct SRSF5 targets. This revealed compensatory mechanisms mediated by SRSF5.

There is a lot of work presented here, and this manuscript certainly has potential. The quality of science is very high, and presentation of data is impeccable. A practical advantage of hGRAD is its 'one-fits-all' nature, where a single plasmid can be

integrated into any cell line that expresses endogenously GFP-tagged RBPs, leading to rapid degradation of the targeted protein. Of course, there is the limitation that this only applies to GFP-tagged proteins. One wonders, whether a different FBP-targeting signal other than targeting GFP could have been tried.

In the opinion of this reviewer, that this manuscript could be published within the 'Tools' section of the Journal, most likely as a shorter story. I think it is not strictly necessary to have Fig 6 with the SRSF5 autoregulation, which distracts from the main methods development character of this study.

Specific comments

- The authors tested the hGRAD system and compare this to osTIR and TRIM21 and others for the degradation of GFP-tagged nuclear RBPs. They expanded this to RBPs in different cellular compartments. It would be interesting to see how these systems compare for the degradation of other type of proteins (non-RBPs) in both nuclear and cytoplasmic compartments. I do not think that there should be intrinsic differences, but since this paper is focused on RBPs, it would be good to have the comparison.

- I may be missing something, but on Fig S1A, why is VHHF not shown in the mCherry panel. According to Fig 2A, it is also transcribed from a dual promoter together with mCherry.

Reviewer #2 (Comments to the Authors (Required)):

This manuscript by Arnold et al describes a system, that they named hGRAD, allowing a rapid and precise depletion of any GFP-tagged proteins in mammalian cells. They successfully adapted a system previously developed in *Drosophila*, that expressed a fusion of a GFP-nanobody and an F-box domain, and is able to bind any GFP-tagged protein in cells to recruit the 26S proteasome for its rapid degradation. Using Western blots and live-cell imaging they showed that this system induces degradation of 5 selected RNA-binding proteins in HeLa cells and pluripotent mouse P19 cells within hours. The comparison with 2 others existing approaches for protein degradation indicates that hGRAD is faster and more efficient. They also show that hGRAD is efficient for targeting RBPs localizing in nuclear condensates, nucleus, and to a lesser extent, the cytoplasm. Moreover, targeting either an externally introduced GFP-tagged version of a protein or introducing a GFP-tag to the endogenous factor by CRISPR/Cas9-mediated knock-in show similar kinetics of degradation. Applying their system *in vivo*, they followed the dynamic changes in the transcriptome using Nascent-Seq within few hours after hGRAD-induced depletion of SRSF5, an RBP with poorly understood function. By integrating iCLIP2 analysis to identify the direct SRSF5 RNA targets with Nascent-seq after acute SRSF5 depletion, they propose that SRSF5 binds and regulates the stability of thousands of mRNAs and lncRNAs.

Overall, the paper is nicely written, well documented and of good quality. They provide a tool that will be of great interest to the scientific community to study the immediate response after inducing a targeted degradation of any protein of interest. One major limitation would be the incompatibility with a tetracycline/doxycycline-inducible transgene expression system. They mostly describe an approach for RNA-binding proteins but hGRAD could be extend to other type of proteins.

Major points:

- There is only one experiment using the proteasome inhibitor MG132 (Figure S3F) which come too late in the results and only appears in supplementary data. Using this treatment as control should come in a first place to validate the efficiency and proteasome-dependency of the system. The authors may want to introduce this data or additional experiments using MG132 in the main figures.
- Overall, while the data showing the efficiency of the hGRAD system are well conducted and convincing, the part characterizing SRSF5 function is less clear.

First, one major concern is the dynamic transcriptome changes observed by Nascent-Seq following SRSF5 depletion with the hGRAD system. The authors claim that the changes are due to SRSF5 depletion while some of the observed effect in the early time points might be indirect due to an inadequate negative control as reference (uninduced T0). Although the expression of the hGRAD system is not impacting cell proliferation, one cannot rule out slight and transient perturbation of the transcriptome in early response to the expression of mCherry and F-Box-VHH-GFP constructs. To prove that the effect is only due to SRSF5 depletion, other negative control(s) should have been used such as: the hGRAD system along with expression of GFP alone or a modified version of the hGRAD system in which the F-box domain would have been mutated or completely removed. The latter would also have been a good control to use in a first place when developing the system since the F-box domain appears critical in the hGRAD system compared to the original DeGraFP system. While it seems difficult to introduce new data including these controls, the author should temper their conclusion in the results/discussion.

Second, the number of SRSF5-binding sites in protein-coding transcripts is relatively high and makes it easy to overlap with populations of low number of transcripts. For example, it is advanced that "51% of the transcripts down-regulated at T2 contained SRSF5 binding sites" (lines 408-409) but this population is around 400 genes while SRSF5 binds 415,245 sites in

more than 8,200 transcripts. The author may want to add statistical tests to determine if these overlaps are more than expected by chance. One suggestion would be to generate cumulative distribution plots (CDPs) after binning SRSF5 targets into high/medium/low/unbound categories and plotting the histogram of abundance changes.

Minor points:

- One could appreciate, in supplementary data or to replace Figure 1B, a sequence alignment of the F-box domain of *Drosophila* Slmb protein along with the human FBPs to justify the choice on FBXW11 among the 70 FBPs.
- Promoting that the hGRAD system works better than other systems is a bit biased since the other systems used in comparison (TRIM21, osTIR1 and aaAFB2) were designed to fit the hGRAD system. The C-terminal fusion (30 kDa) with mCherry and a weak NLS, probably not present in the original (and optimal) systems, could interfere with their activity.
- In Figure S2A the panel with SRPK1-GFP in P19 cells shows already mCherry signal in the no Dox treatment. The authors might comment on this in the figure legends or correct it if it was a mistake.
- Comparing half-life of the same protein in different system (ex: BAC-expressed SRSF5-GFP, half-life of 2.5h in Figure 3C versus SRSF5-endo-GFP, half-life of 2.1h in Figure 5C) need to have statistical test to conclude if a difference of 0.4h is significant or not. If no test has been performed, it is preferable to advance that the half-life is comparable in both constructions.
- The authors may want to double check the references of the Figures in the text, example of wrong reference for Figure 2B in line 294.
- In Figure 6A and in the describing text, it is not clear whether the hour incubation with 4SU is included with the time of Dox treatment or if it is added after the Dox treatment. In the latter, was the Dox absent in the media during the 4SU incubation? If not, then the time point would become 3, 9 and 17 hours. The authors may want to make it clearer in the results and method section.
- Concerning Figure 6D, no information is provided in the figure legend nor the method section regarding the reference(s) transcript(s) used for normalization of the RT-qPCR quantification. Also, the primers used for RT-qPCR are missing in the table S7.
- Title of the results section "SRSF5 binds preferentially to down-regulated transcripts" is misleading and suggest that down-regulation of a transcript could be a determinant for SRSF5 binding. As these "down-regulated transcripts" were previously identified after SRSF5 depletion using the hGRAD system, a better title would be "Down-regulated transcripts after SRSF5 depletion are mostly direct targets of SRSF5"
- It is specified that iCLIP2 were performed with 5 replicates (present in GEO) while only 3 are shown in Figure S5. The authors may want to correct that point or show all replicates. In line with this, one would appreciate to have a Venn diagram showing the overlap of SRSF5 binding sites between the replicates.
- Fig S6A-D is difficult to read and the difference over time is not as obvious as specified in the results section describing different usage of PCE inclusion/exclusion (lines 424 to 431). Simple quantification plots should be added. In line with this, some statistics should be added to conclude if the small differences (5-9%) in PCE inclusion in SRSF3, SRSF6 and SRSF7 transcripts are significant over time.
- The author may want to discuss shortly why the hGRAD system seems to work more efficiently on nuclear rather than cytoplasmic RBPs.

Reviewer #3 (Comments to the Authors (Required)):

This interesting study from the Müller-McNicol lab describes the development of a rapid human protein degradation system, dubbed as hGRAD, for the doxycycline inducible degradation of GFP-tagged proteins through a GFP nanobody fused with the FBXW11 F-box domain. The authors demonstrate that this approach outperforms other existing degron systems in terms of depletion efficiency and that it is suitable for depleting nuclear speckle proteins. Importantly, the authors show that hGRAD is also efficient for the rapid depletion of endogenous SRSF proteins in human and mouse cell lines. Overall, hGRAD represents a new, powerful system for the rapid and facile (assuming the existence of pre-existing GFP-tagged cell lines) depletion of endogenous proteins. However, one remaining concern I have is the extent to which hGRAD has off-target effects. In the last part of the manuscript, the authors have applied hGRAD to the rapid depletion of SRSF5 coupled with a Nascent-Seq time-course experiment in HeLa cells identifying hundreds of genes whose expression is affected by SRSF5 depletion.

Specific comments:

1. My major concern with the presented strategy is lack of exploration of potential off-targets effects of the hGRAD system. This might be particularly important given the surprising gene expression changes elicited by SRSF5 depletion. Using "-" doxycycline as the negative control treatment also prevents expression of the F-box GFP nanobody and thus doesn't allow interrogation of potential off-target degradative events induced by the F-box-nanobody overexpression. Therefore, I would suggest repetition of the Nascent-Seq experiments (at least the most critical time-points) using a WT cell line where SRSF5 is not GFP-tagged while the hGRAD nanobody is induced.
2. I was surprised not to see any splicing analysis of the Nascent-seq data. Given that SRSF5 is a bona-fide splicing regulator, it would be interesting to see how its depletion alters alternative splicing, even if the changes are modest.

Reviewer #1 (Comments to the Authors (Required)):

In this study, Arnold and colleagues developed a novel system for acute depletion of RNA-binding proteins (RBPs), which they termed, hGRAD. This novel system was adapted from the deGradFP system, which was developed to degrade GFP-tagged proteins in *Drosophila*. This degradation system relies on human FBP, FBXW11, a close orthologue to *Drosophila* Slmb, which is fused to an anti-GFP nanobody (VHH-GFP4) and induces degradation of GFP-tagged proteins via recruitment of the endogenous SCF complex.

The authors compare the hGRAD system with other degrading systems, including TRIM21, TIR1 and AFB2, which were cloned in the same vector and was initially tested for the degradation of GFP-SRSF3. The authors extended their analysis to the degradation of another SR protein family member, SRSF5 and to two additional RBPs, SRRM2 and NONO. They concluded that the hGRAD degrades nuclear RBPs more efficiently than the other tested systems.

Next, Arnold and co-authors show that hGRAD does indeed degrade RBPs that localise to nuclear condensates and this system is very efficient across different cell lines and species (HeLa, mouse P19). The authors went on to combine hGRAD-mediated degradation of SRSF5 with Nascent-seq, which due to the rapid degradation of SRSF5 allowed them to identify direct SRSF5 targets. This revealed compensatory mechanisms mediated by SRSF5.

There is a lot of work presented here, and this manuscript certainly has potential. The quality of science is very high, and presentation of data is impeccable. A practical advantage of hGRAD is its 'one-fits-all' nature, where a single plasmid can be integrated into any cell line that expresses endogenously GFP-tagged RBPs, leading to rapid degradation of the targeted protein. Of course, there is the limitation that this only applies to GFP-tagged proteins. One wonders, whether a different FBP-targeting signal other than targeting GFP could have been tried.

We thank the reviewer for the appreciation of our approach. Indeed, our ultimate goal is to use hGRAD with nanobodies against endogenous proteins. But for the moment, there are no nanobodies available for any human RBP. For this reason, we concentrated on the comparison of the different available systems and optimized them for the degradation of mammalian RBPs, which did not work well before. We reasoned that the GFP tag is widely used and well established and that many labs have cell lines already available in which hGRAD can easily be implemented. The beauty of the one-fits-all plasmid is that it can be easily modified to introduce other nanobody sequences or enzymes in the future.

In the opinion of this reviewer, that this manuscript could be published within the 'Tools' section of the Journal, most likely as a shorter story. I think it is not strictly necessary to have Fig 6 with the SRSF5 autoregulation, which distracts from the main methods development character of this study.

We would like to keep this part of the paper as it fulfills the 'proof of principle' requirement for the Tools format of JCB, which is to provide a novel cell biological insight that would not have been possible without the tool. We adjusted the text to ensure that it is not too distractive.

Specific comments

- The authors tested the hGRAD system and compare this to osTIR and TRIM21 and others for the degradation of GFP-tagged nuclear RBPs. They expanded this to RBPs in different cellular compartments. It would be interesting to see how these systems compare for the degradation of other type of proteins (non-RBPs) in both nuclear and cytoplasmic compartments. I do not think that there should be intrinsic differences, but since this paper is focused on RBPs, it would be good to have the comparison.

We agree that the other degradation systems might work better for non-RBPs or RBPs that localize to the cytoplasm. We now mention this possibility in the discussion. We note that we also included one non-RBP in our hGRAD comparison that localizes to the cytoplasm – the kinase SRPK1 – which we worked out clearer in the text now. But we did not compare all four systems with cytoplasmic RBPs and non-RBPs, as it would have been overwhelmingly time-consuming to generate 15 more clonal cell lines for these comparisons.

- I may be missing something, but on Fig S1A, why is VHHF not shown in the mCherry panel. According to Fig 2A, it is also transcribed from a dual promoter together with mCherry.

The slides for hGRAD and the remaining three systems are from two different experiments, in which we used different means to verify the expression and correct localization of the expressed fusion proteins. For aaAFB2/osTIR1/TRIM21 systems, the mCherry is fused to the Ub-ligase and hence mCherry fluorescence was used to verify that the exogenously-expressed Ubiquitin-ligase-mCherry fusions were expressed and localized to the nucleus. In contrast, in the hGRAD system, mCherry is expressed from the same promoter but serves here mainly as an induction reporter, because the hGRAD system uses the endogenous ubiquitin-proteasome system (UPS). To verify that, in the hGRAD system, the GFP nanobody is indeed expressed and nuclear, we performed IF with a specific anti-VHH antibody. We included a note in the legend of the supplementary figure (now Fig. S1B) to clarify this point.

Reviewer #2 (Comments to the Authors (Required)):

This manuscript by Arnold et al describes a system, that they named hGRAD, allowing a rapid and precise depletion of any GFP-tagged proteins in mammalian cells. They successfully adapted a system previously developed in *Drosophila*, that expressed a fusion of a GFP-nanobody and an F-box domain, and is able to bind any GFP-tagged protein in cells to recruit the 26S proteasome for its rapid degradation. Using Western blots and live-cell imaging they showed that this system induces degradation of 5 selected RNA-binding proteins in HeLa cells and pluripotent mouse P19 cells within hours. The comparison with 2 others existing approaches for protein degradation indicates that hGRAD is faster and more efficient. They also show that hGRAD is efficient for targeting RBPs localizing in nuclear condensates, nucleus, and to a lesser extent, the cytoplasm. Moreover, targeting either an externally introduced GFP-tagged version of a protein or introducing a GFP-tag to the endogenous factor by CRISPR/Cas9-mediated knock-in show similar kinetics of degradation. Applying their system *in vivo*, they followed the dynamic changes in the transcriptome using Nascent-Seq within few hours after hGRAD-induced depletion of SRSF5, an RBP with poorly understood function. By integrating iCLIP2 analysis to identify the direct SRSF5 RNA targets with Nascent-seq after acute SRSF5 depletion, they propose that SRSF5 binds and regulates the stability of thousands of mRNAs and lncRNAs.

Overall, the paper is nicely written, well documented and of good quality. They provide a tool that will be of great interest to the scientific community to study the immediate response after inducing a targeted degradation of any protein of interest.

One major limitation would be the incompatibility with a tetracycline/doxycycline-inducible transgene expression system. They mostly describe an approach for RNA-binding proteins but hGRAD could be extended to other type of proteins.

Major points:

- There is only one experiment using the proteasome inhibitor MG132 (Figure S3F) which come too late in the results and only appears in supplementary data. Using this treatment as control should come in a first place to validate the efficiency and proteasome-dependency of

the system. The authors may want to introduce this data or additional experiments using MG132 in the main figures.

We agree with the reviewer and have included another experiment with MG132 and the SRSF3-GFP cell line which we placed at the beginning of the manuscript in new Figures 2E and 2F.

• Overall, while the data showing the efficiency of the hGRAD system are well conducted and convincing, the part characterizing SRSF5 function is less clear. First, one major concern is the dynamic transcriptome changes observed by Nascent-Seq following SRSF5 depletion with the hGRAD system. The authors claim that the changes are due to SRSF5 depletion while some of the observed effect in the early time points might be indirect due to an inadequate negative control as reference (uninduced T0). Although the expression of the hGRAD system is not impacting cell proliferation, one cannot rule out slight and transient perturbation of the transcriptome in early response to the expression of mCherry and F-Box-VHH-GFP constructs. To prove that the effect is only due to SRSF5 depletion, other negative control(s) should have been used such as: the hGRAD system along with expression of GFP alone or a modified version of the hGRAD system in which the F-box domain would have been mutated or completely removed. The latter would also have been a good control to use in a first place when developing the system since the F-box domain appears critical in the hGRAD system compared to the original DeGraFP system. While it seems difficult to introduce new data including these controls, the author should temper their conclusion in the results/discussion.

The reviewer is correct and we regret that we had not included such a control in our Nascent-Seq experiment. We were not able to repeat the Nascent-Seq experiment in time for the revision, but we included a new experiment in which we used the hGRAD master line and the SRSF5 hGRAD line and compared the levels of two SRSF5 target lncRNAs by RT-qPCR. This showed that the downregulation of both lncRNAs upon SRSF5 depletion is not recapitulated as a response to mere hGRAD induction in the control cell lines. The experiment is shown in the new Figure 6F. We also tempered our conclusions in the results section.

Second, the number of SRSF5-binding sites in protein-coding transcripts is relatively high and makes it easy to overlap with populations of low number of transcripts. For example, it is advanced that "51% of the transcripts down-regulated at T2 contained SRSF5 binding sites" (lines 408-409) but this population is around 400 genes while SRSF5 binds 415,245 sites in more than 8,200 transcripts. The author may want to add statistical tests to determine if these overlaps are more than expected by chance. One suggestion would be to generate cumulative distribution plots (CDPs) after binning SRSF5 targets into high/medium/low/unbound categories and plotting the histogram of abundance changes.

We thank the reviewer for these suggestions. We now added the percentage of non-regulated transcripts with SRSF5 binding sites, which is around 15%, in the new Figure S5H. We also separated all expressed transcripts into not bound (no binding sites), lowly bound (lower 20th percentile of binding sites), highly bound (upper 20th percentile of binding sites) and medium bound (remainder of binding sites) and compared their response to SRSF5 depletion (as distributions of \log_2 -transformed fold changes at the different timepoints). We found that at T8 and T16, highly bound transcripts tend to be down-regulated compared to non-bound or lowly bound transcripts. We added these plots in the new Figure 6D and the normalized binding sites in Table S1.

Minor points:

- One could appreciate, in supplementary data or to replace Figure 1B, a sequence alignment of the F-box domain of Drosophila Slmb protein along with the human FBPs to justify the choice on FBXW11 among the 70 FBPs.

We thank the reviewer for the suggestion and added the alignment as new Figure S1A.

- Promoting that the hGRAD system works better than other systems is a bit biased since the other systems used in comparison (TRIM21, osTIR1 and aaAFB2) were designed to fit the hGRAD system. The C-terminal fusion (~30 kDa) with mCherry and a weak NLS, probably not present in the original (and optimal) systems, could interfere with their activity.

We agree with the reviewer that a balanced representation of the results is critical and that we cannot exclude residual biases in our comparison. Regarding the addition of the NLS, we note that we actually took the design of the weak NLS from the publication introducing the aaAFB2 system, which was shown to be more tight and efficient than the osTIR1 system. We now mention this in the text. We assumed that the same weak NLS would also improve degradation of nuclear proteins by osTIR1 and TRIM21.

- In Figure S2A the panel with SRPK1-GFP in P19 cells shows already mCherry signal in the no Dox treatment. The authors might comment on this in the figure legends of correct it if it was a mistake.

We thank the reviewer for spotting this. It was indeed a mistake and we corrected it.

- Comparing half-life of the same protein in different system (ex: BAC-expressed SRSF5-GFP, half-life of 2.5h in Figure 3C versus SRSF5-endo-GFP, half-life of 2.1h in Figure 5C) need to have statistical test to conclude if a difference of 0.4h is significant or not. If no test has been performed, it is preferable to advance that the half-life is comparable in both constructions.

We agree with the reviewer and changed the text accordingly.

- The authors may want to double check the references of the Figures in the text, example of wrong reference for Figure 2B in line 294.

We thank the reviewer for spotting this mistake. We corrected it.

- In Figure 6A and in the describing text, it is not clear whether the hour incubation with 4SU is included with the time of Dox treatment or if it is added after the Dox treatment. In the latter, was the Dox absent in the media during the 4SU incubation? If not, then the time point would become 3, 9 and 17 hours. The authors may want to make it clearer in the results and method section.

We added 4sU after 2 h, 8 h or 16 h of DOX treatment and then left it together with DOX for one hour. The reviewer is hence correct that this would sum up to 3 h, 9 h and 17 h, respectively. We decided to call the timepoints T2, T8 and T16 according to the time where the 4sU was added. We now clarified this in the results and the material and method section.

- Concerning Figure 6D, no information is provided in the figure legend nor the method section regarding the reference(s) transcript(s) used for normalization of the RT-qPCR quantification. Also, the primers used for RT-qPCR are missing in the table S7.

We are thankful to the reviewer for spotting this. We used U6 snRNA for normalization. We now added all used qPCR primers in Table S7.

- Title of the results section "SRSF5 binds preferentially to down-regulated transcripts" is misleading and suggest that down-regulation of a transcript could be a determinant for SRSF5 binding. As these "down-regulated transcripts" where previously identified after

SRSF5 depletion using the hGRAD system, a better title would be "Down-regulated transcripts after SRSF5 depletion are mostly direct targets of SRSF5"

We agree with the reviewer and changed the title.

- It is specified that iCLIP2 were performed with 5 replicates (present in GEO) while only 3 are shown in Figure S5. The authors may want to correct that point or show all replicates. In line with this, one would appreciate to have a Venn diagram showing the overlap of SRSF5 binding sites between the replicates.

We performed iCLIP2 for five SRSF5 replicates and uploaded all them to GEO. Two replicates were loaded on a different gel together with unrelated samples. For representative purposes, we show only the gel with 3 replicates and the corresponding libraries. All replicates were very reproducible. Only binding sites that were found in 4 out of 5 replicates were considered in subsequent analyses. We added this information in the material and method section. For visualization, the UpSet plot below shows how many binding sites were supported by how many replicates (x1-x5). The first 6 bars represent all bindings sites supported by at least 4 replicates, which were used for further analysis.

- Fig S6A-D is difficult to read and the difference over time is not as obvious as specified in the results section describing different usage of PCE inclusion/exclusion (lines 424 to 431). Simple quantification plots should be added. In line with this, some statistics should be added to conclude if the small differences (5-9%) in PCE inclusion in SRSF3, SRSF6 and SRSF7 transcripts are significant over time.

We agree with the reviewer. We now show only Sashimi plots for SRSF5 and SRSF6 and add quantification plots in the new Figure S5J.

- The author may want to discuss shortly why the hGRAD system seems to work more efficiently on nuclear rather than cytoplasmic RBPs.

We do not completely understand the reasons for this, but we added a few thoughts in the discussion.

Reviewer #3 (Comments to the Authors (Required)):

This interesting study from the Müller-McNicoll lab describes the development of a rapid human protein degradation system, dubbed as hGRAD, for the doxycycline inducible degradation of GFP-tagged proteins through a GFP nanobody fused with the FBXW11 F-box domain. The authors demonstrate that this approach outperforms other existing degron systems in terms of depletion efficiency and that it is suitable for depleting nuclear speckle proteins. Importantly, the authors show that hGRAD is also efficient for the rapid depletion of endogenous SRSF proteins in human and mouse cell lines. Overall, hGRAD represents a new, powerful system for the rapid and facile (assuming the existence of pre-existing GFP-tagged cell lines) depletion of endogenous proteins. However, one remaining concern I have is the extent to which hGRAD has off-target effects. In the last part of the manuscript, the authors have applied hGRAD to the rapid depletion of SRSF5 coupled with a Nascent-Seq time-course experiment in HeLa cells identifying hundreds of genes whose expression is affected by SRSF5 depletion.

Specific comments:

1. My major concern with the presented strategy is lack of exploration of potential off-targets effects of the hGRAD system. This might be particularly important given the surprising gene expression changes elicited by SRSF5 depletion. Using "-" doxycycline as the negative control treatment also prevents expression of the F-box GFP nanobody and thus doesn't allow interrogation of potential off-target degradative events induced by the F-box-nanobody overexpression. Therefore, I would suggest repetition of the Nascent-Seq experiments (at least the most critical time-points) using a WT cell line where SRSF5 is not GFP-tagged while the hGRAD nanobody is induced.

The reviewer is correct and we regret that we had not include such a control in our Nascent-Seq experiment. We were not able to repeat the Nascent-Seq experiment in time for the revision, but we included a new experiment in which we used the hGRAD master line and the SRSF5 hGRAD line and compared the levels of two SRSF5 target lncRNAs by RT-qPCR. The reduction in both transcripts can be seen when SRSF5 is depleted but not in control cells where only the hGRAD system is induced. The experiment is shown in the new Figure 6F.

2. I was surprised not to see any splicing analysis of the Nascent-seq data. Given that SRSF5 is a bona-fide splicing regulator, it would be interesting to see how its depletion alters alternative splicing, even if the changes are modest.

We have analyzed the Nascent-Seq data for intron retention using IRFinder and added the numbers in the text. We now performed additional splicing analyses with MAJIQ and detected only modest splicing changes, which were mostly transient. The MAJIQ results have been added to Figure S5K.

October 13, 2023

RE: JCB Manuscript #202304030R

Prof. Michaela Müller-McNicoll
Goethe University Frankfurt
Institute for Molecular Biosciences
Max-von-Laue-Str. 13
Frankfurt am Main 60438
Germany

Dear Prof. Müller-McNicoll:

Thank you for submitting your revised manuscript entitled "hGRAD - a versatile 'one-fits-all' system to acutely deplete RNA binding proteins from condensates". The paper has now been seen again by two of the original reviewers. As you can see from the appended comments, the reviewers are now supportive of publication after you consider the comments of Reviewer #2. Therefore, we would be happy to publish your paper in JCB pending final revisions necessary to meet our formatting guidelines (see details below).

****Please address the specific remaining concerns of reviewer #2 directly. Concerning the request for further analysis, although we agree that such information would be useful if it is practical to provide it promptly, we leave the decision up to you whether to provide such an addition. Please be sure to provide a point-by-point rebuttal to these final reviewer comments.****

A. MANUSCRIPT ORGANIZATION AND FORMATTING:

1) Text limits: Character count for Tools is < 40,000, not including spaces. Count includes the abstract, introduction, results, discussion, and acknowledgments. Count does not include title page, materials and methods, figure legends, references, tables, or supplemental legends. You are currently below this limit but if you need to add more text to the paper during revision, please try to be as concise as possible.

2) Figure formatting: Scale bars must be present on all microscopy images, including inset magnifications. Molecular weight or nucleic acid size markers must be included on all gel electrophoresis.

****Please provide molecular weight markers for the blots in figures 3C, 5C and S5A.****

3) Statistical analysis: Error bars on graphic representations of numerical data must be clearly described in the figure legend. The number of independent data points (n) represented in a graph must be indicated in the legend. Statistical methods should be explained in full in the materials and methods. For figures presenting pooled data the statistical measure should be defined in the figure legends. In your methods, you currently state: "GraphPad Prism was used for graphics/statistics." You must indicate what statistical tests, if any, were used in each of your experiments (both in the figure legend itself and in a separate methods section) as well as the parameters of the test (for example, if you ran a t-test, please indicate if it was one- or two-sided, etc.). Also, if you used parametric tests, please indicate if the data distribution was tested for normality (and if so, how). If not, you must state something to the effect that "Data distribution was assumed to be normal but this was not formally tested."

4) Materials and methods: Should be comprehensive and not simply reference a previous publication for details on how an experiment was performed. Please provide full descriptions (at least in brief) in the text for readers who may not have access to referenced manuscripts. The text should not refer to methods "...as previously described."

5) Please be sure to provide the sequences for all of your primers/oligos and RNAi constructs in the materials and methods. You must also indicate in the methods the source, species, and catalog numbers (where appropriate) for all of your antibodies.

6) Microscope image acquisition: The following information must be provided about the acquisition and processing of images:

- a. Make and model of microscope
- b. Type, magnification, and numerical aperture of the objective lenses
- c. Temperature
- d. imaging medium
- e. Fluorochromes

f. Camera make and model

g. Acquisition software

h. Any software used for image processing subsequent to data acquisition. Please include details and types of operations involved (e.g., type of deconvolution, 3D reconstitutions, surface or volume rendering, gamma adjustments, etc.).

7) References: There is no limit to the number of references cited in a manuscript. References should be cited parenthetically in the text by author and year of publication. Abbreviate the names of journals according to PubMed.

**Please note that we do not allow any "supplemental references". Therefore, you will need to remove this list from the supplementary material and add any non-duplicated references to the main reference list.*

8) Supplemental materials: There are normally strict limits on the allowable amount of supplemental data. Tools may have up to 5 supplemental figures. At the moment, you meet this limit but if you need to add to the total in order to address reviewer #2's final comments, we can allow an extra figure in this case.

Please also note that tables, like figures, should be provided as individual, editable files. A summary of all supplemental material (that is, in addition to the supplementary figure legends) should appear at the end of the Materials and methods section. Please see any recent JCB paper for an example of this.

9) eTOC summary: A ~40-50 word summary that describes the context and significance of the findings for a general readership should be included on the title page. The statement should be written in the present tense and refer to the work in the third person. It should contain "First author name(s) et al..." to match our preferred style.

10) Conflict of interest statement: JCB requires inclusion of a statement in the acknowledgements regarding competing financial interests. If no competing financial interests exist, please include the following statement: "The authors declare no competing financial interests." If competing interests are declared, please follow your statement of these competing interests with the following statement: "The authors declare no further competing financial interests."

11) A separate author contribution section is required following the Acknowledgments in all research manuscripts. All authors should be mentioned and designated by their first and middle initials and full surnames. We encourage use of the CRediT nomenclature (<https://casrai.org/credit/>).

12) ORCID IDs: ORCID IDs are unique identifiers allowing researchers to create a record of their various scholarly contributions in a single place. Please note that ORCID IDs are now *required* for all authors. At resubmission of your final files, please be sure to provide your ORCID ID and those of all co-authors.

13) As you know, JCB now requires authors to submit Source Data used to generate figures containing gels and Western blots with all revised manuscripts. Thank you for providing Source Data for most of your blots and gels. However, there are two items that require your attention. First, the uncropped blots for figure 2E and the uncropped gel for figure S3B appear to be missing. Second, you must provide one Source Data file for each figure that contains gels and/or blots along with your revised manuscript files (not one per figure panel). The files can be multiple pages, if necessary, but each figure should have only a single PDF file (e.g. the Source Data for the blots in Figure 2B, D, and E should all be placed into a single PDF file).

You should endeavor to retain a minimum resolution of 300 dpi or pixels per inch. Please review our instructions for export from Photoshop, Illustrator, and PowerPoint here: <https://rupress.org/jcb/pages/submission-guidelines#revised>

14) Journal of Cell Biology now requires a data availability statement for all research article submissions. These statements will be published in the article directly above the Acknowledgments. The statement should address all data underlying the research presented in the manuscript.

While we appreciate that you've provided the accession numbers for the sequencing data, you must also indicate how the data supporting all of the other figures in the paper will be made available. Please visit the JCB instructions for authors for guidelines and examples of statements at (<https://rupress.org/jcb/pages/editorial-policies#data-availability-statement>).

If you do not provide all the data in the paper itself (including replicates), though, then consider adding a statement such as: "All other data are available from the corresponding author upon reasonable request."

B. FINAL FILES:

-- Cover images: If you have any striking images related to this story, we would be happy to consider them for inclusion on the journal cover. Submitted images may also be chosen for highlighting on the journal table of contents or JCB homepage carousel.

Images should be uploaded as TIFF or EPS files and must be at least 300 dpi resolution.

****It is JCB policy that if requested, original data images must be made available to the editors. Failure to provide original images upon request will result in unavoidable delays in publication. Please ensure that you have access to all original data images prior to final submission.****

****The license to publish form must be signed before your manuscript can be sent to production. A link to the electronic license to publish form will be sent to the corresponding author only. Please take a moment to check your funder requirements before choosing the appropriate license.****

Thank you for your attention to these final processing requirements. Please revise and format the manuscript and upload materials within 7-14 days. If complications arising from measures taken to prevent the spread of COVID-19 will prevent you from meeting this deadline (e.g. if you cannot retrieve necessary files from your laboratory, etc.), please let us know and we can work with you to determine a suitable revision period.

Thank you for this interesting contribution, we look forward to publishing your paper in Journal of Cell Biology.

Sincerely,

Kenneth Yamada, MD, PhD
Senior Editor
Journal of Cell Biology

Tim Spencer, PhD
Executive Editor
Journal of Cell Biology

Reviewer #2 (Comments to the Authors (Required)):

The authors engaged comprehensively with all of the reviewers' comments. I have no concerns and support publication.

Reviewer #3 (Comments to the Authors (Required)):

In the revised manuscript, the authors have addressed most of the comments raised by the reviewers. However, the major concern of this reviewer (as well as of reviewer 2), namely the assessment of hGRAD specificity, has not been sufficiently addressed. Due to lack of sufficient time provided by the journal, the authors didn't perform the suggested Nascent-Seq experiments but only tested the expression of two genes by focused RT-PCR assays (perhaps the JCB policies could change allowing flexibility for the performance of key experiments). However, I would expect to see at least 10-20 genes so that more accurate conclusions could be drawn. In any case, I still believe that this study describes the development of a tool with great potential (assuming that the specificity will not be a problem in the future) for the facile and acute depletion of nuclear proteins and could thus be of interest to the scientific community.

I would suggest incorporation the following minor (but necessary) statement to the abstract to address the lack of appropriate experimentation for assessing hGRAD specificity: "Future work is required to assess potential off-target effects of hGRAD."

I would also encourage the authors to provide a little more information about the splicing analysis in Figure S5. For example, the overlap of affected introns between the different time points as well as an analysis assessing if the regulated introns are bound by SRSF5 could be presented. Whatever the conclusion of these analyses might be, it may be useful for the reader's interpretation of the data.

Response to the Reviewers

Reviewer #3 (Comments to the Authors (Required)):

In the revised manuscript, the authors have addressed most of the comments raised by the reviewers. However, the major concern of this reviewer (as well as of reviewer 2), namely the assessment of hGRAD specificity, has not been sufficiently addressed. Due to lack of sufficient time provided by the journal, the authors didn't perform the suggested Nascent-Seq experiments but only tested the expression of two genes by focused RT-PCR assays (perhaps the JCB policies could change allowing flexibility for the performance of key experiments). However, I would expect to see at least 10-20 genes so that more accurate conclusions could be drawn. In any case, I still believe that this study describes the development of a tool with great potential (assuming that the specificity will not be a problem in the future) for the facile and acute depletion of nuclear proteins and could thus be of interest to the scientific community.

I would suggest incorporation the following minor (but necessary) statement to the abstract to address the lack of appropriate experimentation for assessing hGRAD specificity: "Future work is required to assess potential off-target effects of hGRAD."

We agree with the reviewer and added this sentence at the end of our conclusion.

I would also encourage the authors to provide a little more information about the splicing analysis in Figure S5. For example, the overlap of affected introns between the different time points as well as an analysis assessing if the regulated introns are bound by SRSF5 could be presented. Whatever the conclusion of these analyses might be, it may be useful for the reader's interpretation of the data.

We added two panels in Figure S5 to show the dynamic regulation of intron retention and the overlap of regulated AS events between the three timepoints.